# Amotl2a interacts with the Hippo effector Yap1 and the Wnt/β-catenin effector Lef1 to control tissue size in zebrafish

**Sobhika Agarwala[1,2], Sandra Duquesne[1,2], Kun Liu[1,3], Anton Boehm[1,3], Lin Grimm[2], Sandra Link[1,2], Sabine König[1], Stefan Eimer[1,4,5], Olaf Ronneberger[1,3], Virginie Lecaudey[2]\*†**

[1]BIOSS Centre for Biological Signalling Studies, Albert Ludwigs University of Freiburg, Freiburg im Breisgau, Germany; [2]Developmental Biology, Institute for Biology I, Faculty of Biology, Albert Ludwigs University of Freiburg, Freiburg im Breisgau, Germany; [3]Image Analysis Lab, Institute for Computer Science, Albert Ludwigs University of Freiburg, Freiburg im Breisgau, Germany; [4]ZBSA Center for Biological Systems Analysis, Albert Ludwigs University of Freiburg, Freiburg im Breisgau, Germany; [5]Institute for Biology III, Faculty of Biology, Albert Ludwigs University of Freiburg, Freiburg im Breisgau, Germany

**\*For correspondence:**
Lecaudey@bio.uni-frankfurt.de

**Present address:** †BIOSS Centre for Biological Signalling Studies, Albert Ludwigs University of Freiburg, Freiburg im Breisgau, Germany

**Competing interests:** The authors declare that no competing interests exist.

**Reviewing editor**: Tanya T Whitfield, University of Sheffield, United Kingdom

**Abstract** During development, proliferation must be tightly controlled for organs to reach their appropriate size. While the Hippo signaling pathway plays a major role in organ growth control, how it senses and responds to increased cell density is still unclear. In this study, we use the zebrafish lateral line primordium (LLP), a group of migrating epithelial cells that form sensory organs, to understand how tissue growth is controlled during organ formation. Loss of the cell junction-associated Motin protein Amotl2a leads to overproliferation and bigger LLP, affecting the final pattern of sensory organs. Amotl2a function in the LLP is mediated together by the Hippo pathway effector Yap1 and the Wnt/β-catenin effector Lef1. Our results implicate for the first time the Hippo pathway in size regulation in the LL system. We further provide evidence that the Hippo/Motin interaction is essential to limit tissue size during development.

## Introduction

Control of cell number is a critical process during the development of an organism. While proliferation is essential for organs to reach a correct size, failure to tightly regulate proliferation can lead to organ overgrowth and tumor formation. Proliferation must therefore be tightly controlled and coordinated with other developmental processes so that organs reach their proper final size, but do not exceed it. How this is achieved, however, is still largely unknown.

In the past 15 years, the Hippo signaling pathway has been identified as a major regulator of organ size during development and homeostasis, by promoting cell death and differentiation and inhibiting proliferation. First identified in *Drosophila* (*Pan, 2007*), the Hippo signaling pathway is highly conserved in vertebrates (*Halder and Johnson, 2010*; *Pan, 2010*). When Hippo signaling is active, the Hippo pathway effectors YAP1 (Yes-associated protein 1) and TAZ (transcriptional co-activator with a PDZ domain), the vertebrate homologs of the *Drosophila* Yorkie, are phosphorylated by a cascade of kinases leading to their sequestration in the cytoplasm and/or their degradation. In contrast, when the Hippo signaling pathway is inactive, YAP/TAZ can translocate into the nucleus and mediate transcription of genes that promote proliferation and inhibit apoptosis (*Zhao et al., 2011*; *Barry and Camargo, 2013*; *Yu and Guan, 2013*; *Bossuyt et al., 2014*).

**eLife digest** How do organs and tissues know when to stop growing? A cell communication pathway known as Hippo signaling plays a central role as it can tell cells to stop dividing. It is activated when cells in developing tissues come into contact with each other and causes a protein called Yap1 to be modified, which prevents it from entering the cell nucleus to activate genes that are involved in cell division.

In a zebrafish embryo, an organ called the lateral line forms from a cluster of cells that migrate along the embryo's length. At regular intervals, the cluster deposits small bunches of cells from its trailing end. The resulting loss of cells from the cluster is balanced by cell division at the front of the cluster, which is triggered by another signaling pathway called Wnt signaling. A protein of the 'Motin' family called Amotl2a is present in this migrating cluster. Motin proteins form junctions between cells and inhibit the activity of Yap1, but it is not known whether they are involved in regulating the size of organs.

Here, Agarwala et al. used the lateral line as a model to study the control of organ size in zebrafish embryos. The experiments show that when Amotl2a is absent, the migrating cell cluster becomes larger, with the highest levels of cell division occurring at its trailing end. Yap1 and a protein involved in Wnt signaling called Lef1 are also present in the cluster and are required for it to be normal in size. In zebrafish that lack Amotl2a, the additional loss of Yap1 prevents this cluster from becoming too large. From these and other results, it appears that Amotl2a regulates the size of the lateral line cell cluster by restricting the ability of Yap1 and Lef1 to promote cell division.

Agarwala et al.'s findings demonstrate a role for Amotl2a in controlling the size of organs. A future challenge is to understand the details of how it restricts the activities of Yap1 and Lef1.

Contact inhibition of proliferation was found to be largely mediated by the Hippo signaling pathway (*Zhao et al., 2007*). Downstream of cell–cell adhesion and apicobasal polarity, many junction-associated proteins including E-cadherin, α-catenin, and proteins of the Crumbs and Par complexes promote Hippo signaling (*Kim et al., 2011*; *Schlegelmilch et al., 2011*; *Silvis et al., 2011*; *Enderle and McNeill, 2013*). These proteins serve as scaffold for the Hippo pathway kinases MST1/2 and LATS1/2 leading to YAP/TAZ phosphorylation and retention in the cytoplasm or degradation (*Grusche et al., 2010*; *Boggiano and Fehon, 2012*; *Irvine, 2012*; *Schroeder and Halder, 2012*; *Gumbiner and Kim, 2014*).

Recent studies have shown that, in addition to changes in cell density, YAP/TAZ responds to changes in cell shape, tension forces, and substrate stiffness. This, however, seems largely independent of the canonical Hippo kinase cascade but depends on actin (*Dupont et al., 2011*; *Wada et al., 2011*; *Halder et al., 2012*; *Aragona et al., 2013*). The actin cytoskeleton indeed plays an important role in integrating and transmitting upstream signals to the Hippo pathway effectors YAP and TAZ (*Gaspar and Tapon, 2014*). Yet, how this is achieved is not well understood.

Several recent reports suggest that the Motin family of junction-associated proteins could play a central role here. AMOT, AMOTL1, and AMOTL2 are scaffold proteins associated with tight-junctions, required for tight junction integrity (*Bratt et al., 2002*; *Sugihara-Mizuno et al., 2007*; *Zheng et al., 2009*) and endothelial cell migration (*Troyanovsky et al., 2001*; *Bratt et al., 2005*; *Aase et al., 2007*; *Wang et al., 2011*; *Hultin et al., 2014*; *Moleirinho et al., 2014*). Recently, Motin proteins have further been shown to interact with YAP and TAZ via their PPxY motifs and the WW motifs of YAP and TAZ (*Chan et al., 2011*; *Wang et al., 2011*; *Zhao et al., 2011*; *Hirate et al., 2013*; *Hong, 2013*; *Lucci et al., 2013*; *Yi et al., 2013*). In most cases, this physical interaction leads to the inhibition of YAP/TAZ via cytoplasmic retention, similar to, but independent of the canonical Hippo pathway. Interestingly, there is a competition between YAP/TAZ and F-actin to bind to Motin proteins (*Mana-Capelli et al., 2014*). Motin proteins have thus been proposed to mediate the response of YAP/TAZ to changes in the actin cytoskeleton downstream of mechanical signals. Yet, whether Motin proteins play such a central role in regulating organ growth in vivo in developing organisms is still largely unknown.

The posterior lateral line (pLL) system in zebrafish provides an excellent model system to address this question. The pLL is a sensory system comprised of mechanosensory organs, the neuromasts,

scattered on the surface of the body. The pLL primordium (pLLP) consists of about 100 progenitors that delaminate from a cranial placode and migrate posteriorly towards the tip of the tail (*Metcalfe et al., 1985*; *Ghysen and Dambly-Chaudiere, 2004*). As the pLLP migrates, small groups of cells within its trailing region undergo cell shape changes to assemble into rosette-like structures, called proneuromasts. This assembly requires Fibroblast Growth Factor (FGF) signaling and the downstream effectors Shroom3 and Rock2a to activate non-muscle myosin and apical-constriction (*Lecaudey et al., 2008*; *Nechiporuk and Raible, 2008*; *Ernst et al., 2012*; *Harding and Nechiporuk, 2012*). Once assembled, proneuromasts are deposited behind the migrating pLLP and differentiate into functional neuromasts.

Proneuromast deposition comes with a significant loss of cells for the migrating pLLP. Wnt/β-catenin signaling partially compensates for this loss by promoting proliferation in the leading region (*Gamba et al., 2010*; *Aman et al., 2011*; *McGraw et al., 2011*; *Valdivia et al., 2011*; *Matsuda et al., 2013*). Hereafter, we use the term 'leading region' for the cells in the posterior part of the pLLP that do not assemble into rosettes. In contrast, with 'trailing region', we refer to the part of the pLLP where cells change their shape to assemble into rosettes and where FGF signaling is active.

Here, we use the pLLP to address the mechanisms required to control proliferation and tissue size during organ development. We focus on the role of the Motin protein Angiomotin-like 2a (Amotl2a) that has been reported to be expressed in the migrating pLLP, to be a target of FGF signaling (*Huang et al., 2007*), to inhibit Wnt/β-catenin signaling during early zebrafish development (*Li et al., 2012*), and is a potential candidate to interact with the Hippo signaling pathway. Loss of Amotl2a function results in a significant increase in pLLP size due to overproliferation. In a yeast two-hybrid (Y2H) screen, we identified the zebrafish Hippo pathway effectors Yap1 and Taz as strong interacting partners of Amotl2a. We show that Yap1 is required for the pLLP to reach its correct size and that reducing the level of Yap1 suppresses the overproliferation in *amotl2a* mutant pLLP. Finally we show that, in addition to Yap1, the Wnt/β-catenin pathway effector Lef1 also mediates the hyperplasia phenotype of *amotl2a* mutants. This leads us to propose that Amotl2a, possibly by forming a ternary complex with Yap1 and β-catenin, limits proliferation in the trailing part of the pLLP. Here, we report the first mechanism that limits proliferation in the pLLP. In addition, we implicate the Hippo effector Yap1 in size control in the pLLP for the first time. Altogether, our results strongly suggest that Motin proteins play a major role in limiting organ growth during development by negatively regulating Yap/Taz and Wnt/β-catenin activities in vivo.

## Results

### *amotl2a* is expressed in the pLLP and localizes at cell–cell junctions

*amotl2a* has been shown to be expressed in the pLLP (*Huang et al., 2007*). By combining in situ hybridization (ISH) with Green Fluorescent Protein (GFP) immunostaining in *cldnb:gfp* transgenic embryos, we showed that *amotl2a* was expressed throughout migration (*Figure 1A–C*) in most of the pLLP with reduced or no expression in the most leading and trailing domains (*Figure 1D–I*). The other *motin* genes *amot*, *amotl1*, and *amotl2b* were not expressed in the pLLP (*Figure 1—figure supplement 1A–C*). To determine the intracellular localization of Amotl2a in the pLLP, we fused Amotl2a to the red fluorescent protein TdTomato (TdT). Amotl2a-TdT was present in the cytoplasm and was strongly concentrated at the most apical, constricted part of the cells in assembling proneuromasts (*Figure 1J–K'*). In addition, it appeared as an apical ring in deposited neuromasts (*Figure 1L,L'*).

We next asked if *amotl2a* expression in the pLLP also required FGF signaling as it has been reported during early development (*Huang et al., 2007*). *amotl2a* expression was specifically lost in the pLLP of *fgf3$^{-/-}$;fgf10$^{-/-}$* double mutants (*Figure 1M–R*), upon expression of a dominant negative FGFR1 receptor (*Figure 1—figure supplement 1D–I*), and in embryos treated with the FGFR inhibitor SU5402 (not shown). Conversely, ectopic activation of FGF signaling using the *Tg(hsp70l:fgf3-Myc)$^{zf115}$* transgenic line led to an expansion of the *amotl2a* expression domain in the pLLP as compared to controls (*Figure 1S–X* and *Figure 1—figure supplement 1J,K*). These results indicated that FGF signaling is necessary and sufficient for *amotl2a* expression in the pLLP.

### Amotl2a is not essential for tight junction and proneuromast assembly in the pLLP

To investigate the function of Amotl2a in the pLLP, we used a previously published translation-blocking morpholino (Amotl2aMo) (*Huang et al., 2007*; *Wang et al., 2011*; *Li et al., 2012*). At low

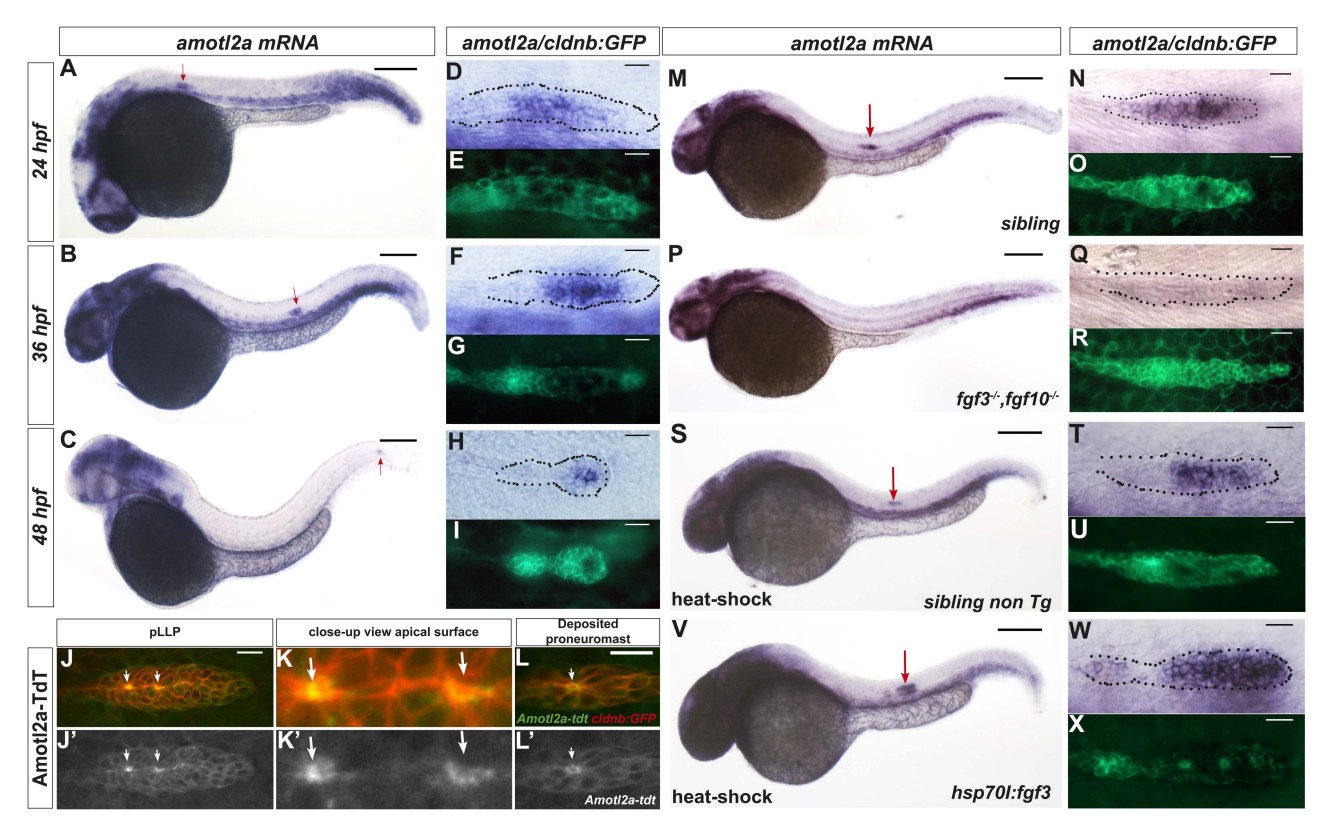

**Figure 1**. *amotl2a* is expressed in the pLLP and localizes at the cell apical side. (**A–I**) *cldnb:gfp* embryos stained with an *amotl2a* antisense RNA probe and an anti-GFP antibody (**E**, **G**, **I**) at the indicated stages. Red arrows indicate the posterior lateral line primordium (pLLP). (**D–I**) Close-up views of the pLLP. (**J–L'**) Maximum intensity projection (MIP) of Z-stacks of the pLLP (**J–J'**) and a recently deposited neuromast (**L–L'**) in *cldnb:gfp* embryos injected with *amotl2a-TdT* mRNA. (**K–K'**) Close-up views of (**J–J'**). White arrows indicate rosette centers. Colors have been inverted. (**M–X**) 30 hpf *cldnb:gfp* embryos stained with an *amotl2a* in situ hybridization (ISH) probe and an anti-GFP antibody (**O**, **R**, **U**, **X**) in the indicated genetic background. The right column shows the primordium at higher magnification. In all figures, scale bars correspond to 200 µm for whole-embryo images and 20 µm in close up views of the pLLP. In all figures, n is the total number of embryos/primordia analysed and N is the number of biological replicates (*Figure 1—figure supplement 1*, *Figure 1—source data 1*).

The following source data and figure supplement are available for figure 1:

**Source data 1**. Relative length of the *amotl2a*-free (Excel sheet 1 related to panel J) and *amotl2a*-expressing domain (Excel sheet related to panel K).

**Figure supplement 1**. Only *amotl2a* is expressed in the pLLP and its expression is controlled by FGF signaling.

doses, Amotl2aMo did not lead to any obvious morphological defects (*Figure 2—figure supplement 1A–D*) (*Wang et al., 2011*) and efficiently blocked the translation of a fusion between the ATG region of *amotl2a* and GFP (MoBS_Amotl2a-GFP, *Figure 2—figure supplement 1E,F*).

Since Motin proteins are necessary for tight junction integrity in several contexts (*Bratt et al., 2002*; *Sugihara-Mizuno et al., 2007*; *Zheng et al., 2009*), we performed ZO1 and phalloidin staining to label tight junctions and actin, respectively. While mature rosettes had similarly strong apical actin and ZO1 staining in morphant and control embryos (white arrows in *Figure 2C–F*), the distance between the tip of the pLLP and the most recent fully assembled rosette (yellow arrows), appeared increased in *amotl2a* morphants (double arrowhead in *Figure 2A,B*). Yet, neuromasts were deposited normally (*Figure 2G,H*). This result suggested that rosette assembly was slightly delayed in *amotl2a* morphants but that Amotl2a was neither essential for tight junction assembly nor for neuromast formation and deposition.

In addition, the pLLP of *amotl2a* morphant embryos migrated 35% slower than controls (*Figure 2I–M*). Yet, all primordia reached the tip of the tail (*Figure 2G,H*). Expression of the two

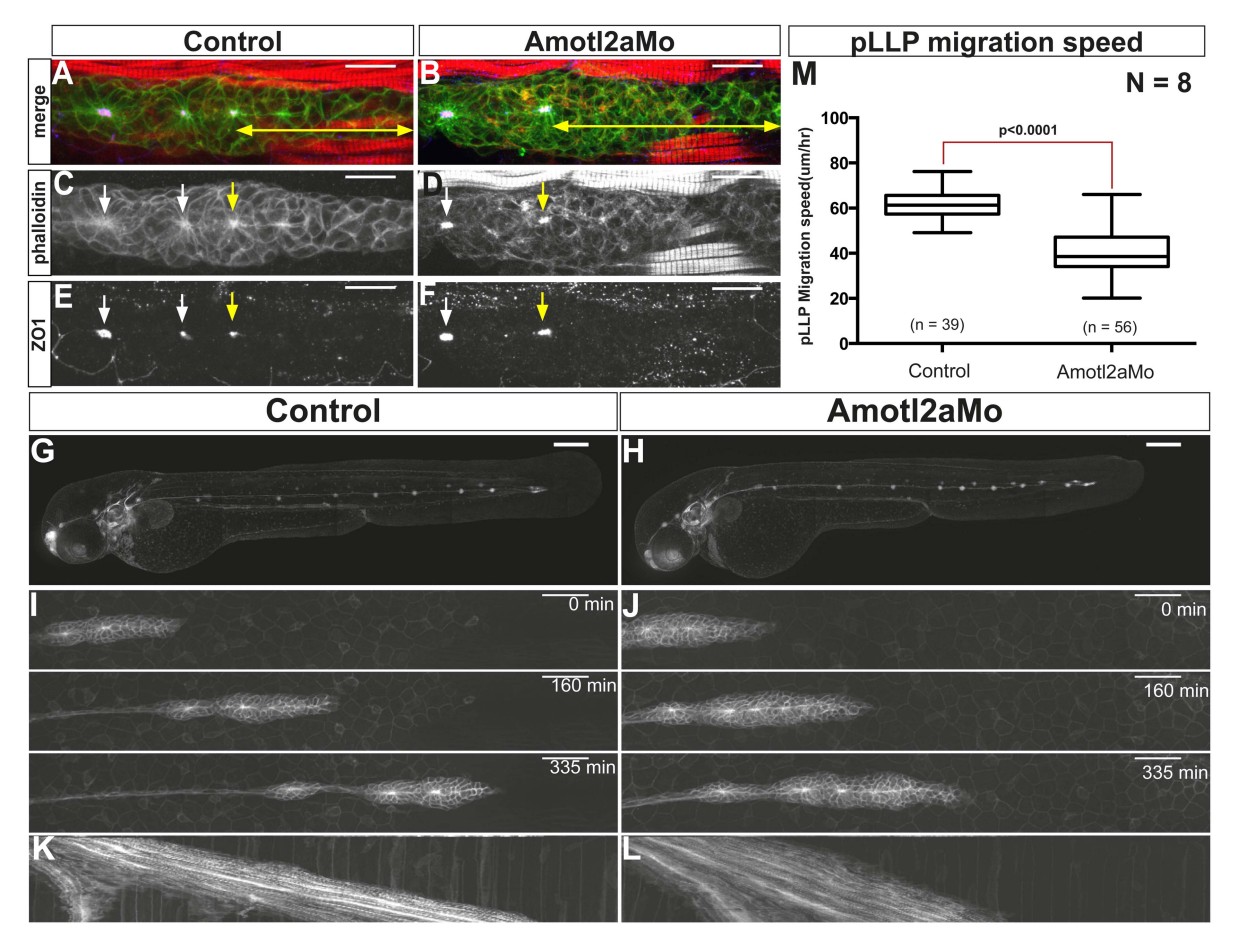

**Figure 2**. Amotl2a is not essential for proneuromast assembly but for proper migration. (A–F) MIP of Z-stacks of pLLP stained with ZO-1 (blue) and GFP (green) antibodies and phalloidin (red) in control (A, C, E) and *amotl2a* morphant (B, D, F) *cldnb:gfp* embryos at 30 hpf. (G, H) MIP of overview images of control (G) and *amotl2a* morphant (H) embryos after completion of migration. (I, J) Snapshots of time-lapse movies at the indicated timepoints showing a delay in migration in *amotl2a* morphants (J) as compared to controls (I). (K, L) Corresponding kymographs used to measure the migration speed. (M) Boxplot comparing the migration speeds (*Figure 2—source data 1*, *Figure 2—figure supplements 1, 2*).

The following source data and figure supplements are available for figure 2:

**Source data 1**. Migration speed in *amotl2a* morphants.

**Figure supplement 1**. Amotl2aMo efficiency.

**Figure supplement 2**. *cxcr4b* and *cxcr7b* expression are not affected in *amotl2a* morphants.

G-protein-coupled receptors *cxcr4b* and *cxcr7b* was not detectably affected in morphants (*Figure 2—figure supplement 2*), suggesting that a deregulation of their expression is unlikely to account for the decrease in migration speed. Altogether, these results suggested that while rosette assembly and migration are delayed in *amotl2a* morphants, the pLLP deposits neuromasts and eventually migrates to the tip of the tail.

## Amotl2a is essential to restrict the number of cells in the pLLP, and thus to limit its size

Interestingly, the size of *amotl2a* morphant pLLP appeared obviously increased throughout the migration process (*Figure 3A–C*, also *Figure 2I,J*). To determine whether this was due to an increase in cell number, we developed an algorithm to automatically count cells in the pLLP based on the

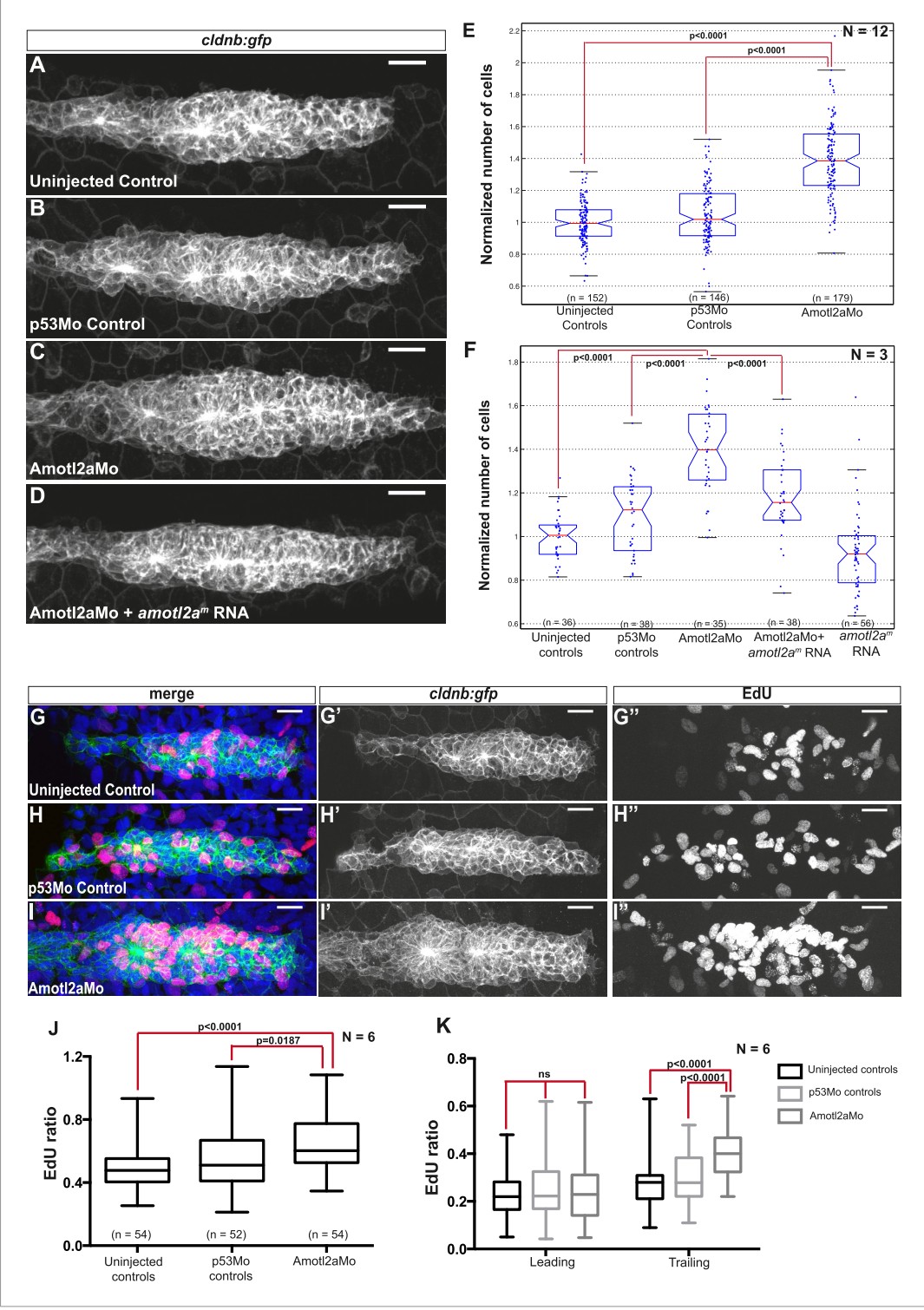

**Figure 3**. Amotl2a is required to limit proliferation in the pLLP. (**A–D**) MIP of Z-stacks of pLLP in *cldnb:gfp* embryos injected as indicated. (**E**, **F**) Boxplots showing the number of cells in the primordia of indicated groups, normalized to the control group. (**G–I''**) MIP of Z-stacks of pLLP in *cldnb:gfp* embryos stained with EdU and DAPI showing the green (membranes, middle), red (EdU, right), and merge (left) channels. (**J**, **K**) Boxplot showing the comparison of the EdU index in the indicated experimental conditions in whole primordia (**J**) or separately in the leading and trailing region (**K**) (*Figure 3—source data 1–3*, *Figure 3—figure supplements 1, 2*, *Figure 3—source data 4*).

*Figure 3. continued on next page*

*Figure 3. Continued*

The following source data and figure supplements are available for figure 3:

**Source data 1**. Cell counts in *amotl2a* morphants.

**Source data 2**. Cell counts in rescue experiment of *amotl2a* morphants.

**Source data 3**. EdU ratio in the entire pLLP (Excel sheet 1 related to panel J), leading region and trailing region (Excel sheet 2 related to panel K) of *amotl2a* morphants.

**Source data 4**. Cell counts in *amotl2a*-overexpressing embryos.

**Figure supplement 1**. Successive steps of the automated 'cell-counting' algorithm.

**Figure supplement 2**. Overexpression of *amotl2a* leads to reduced cell number in the pLLP.

membrane labeling in *cldnb:gfp* embryos (see 'Materials and methods' and *Figure 3—figure supplement 1*). Cell counts were done in pLLP that had reached the middle of the yolk extension in both controls and *amotl2a* morphants (thereafter referred to as 'mid-migration', red arrowhead in *Figure 2—figure supplement 1,* see also 'Materials and methods'). Cell counts indicated that the number of cells in *amotl2a* morphant pLLP was increased by 35–38% as compared to either uninjected or p53Mo-injected embryos (*Figure 3E*, p = 1.58E-40 and p = 9.53E-31, respectively). To confirm the specificity of the morphant phenotype, we co-injected Amotl2aMo with *amotl2a^m*, an RNA that was insensitive to the morpholino. Co-injection of Amotl2aMo with *amotl2a^m* RNA partially rescued the morphant phenotype (*Figure 3D,F*). While overexpression of *amotl2a* at the concentration used for the rescue experiment did not significantly reduce the number of cells in the pLLP (*Figure 3F*), injection at a higher concentration induced a moderate, but significant decrease in cell number (*Figure 3—figure supplement 2*, −15%, p = 5.20E-07). Altogether, these results indicated that Amotl2a was essential to limit the number of cells in the migrating pLLP.

## Amotl2a is required to limit the proliferation rate in the trailing region of the pLLP

To determine the cause of the increase in cell counts in the pLLP of *amotl2a* morphants, we quantified proliferation rates using EdU to label proliferating cells (*Figure 3G–I″*). There was a significant increase in the EdU index of about 15% in the pLLP of *amotl2a* morphants as compared to controls (*Figure 3J*, p < 0.05). Since *amotl2a* was not expressed in the most leading part of the pLLP, we determined the proliferation rates separately in the leading vs trailing region of the primordium. While there was no difference in the proliferation index in the leading region between morphants and controls, there was a 34% increase in the trailing region (*Figure 3K*, p < 0.0001). Altogether, these results indicated that Amotl2a is required to restrict the number of cells in the pLLP by limiting proliferation in the region where rosettes assemble.

## TALEN-induced non-sense mutations in *amotl2a* phenocopy the *amotl2a* knock-down phenotype

To confirm the specificity of the Amotl2Mo-induced phenotype, we generated *amotl2a* mutants using the transcription activator-like effector nuclease (TALEN) technique. Among several identified *amotl2a* alleles, two were further analyzed that introduced a frame-shift and a premature STOP codon in the third exon (*Figure 4A*). The putative resulting truncated proteins would lack most of the coiled-coil domain and the PDZ-binding domain (*Figure 4B*). Both alleles had identical phenotypes. *amotl2a^{−/−}* homozygous mutants were morphologically indistinguishable from their siblings (*Figure 4C–F*) but had larger primordia (*Figure 4G,H*). Cell count analyses revealed a significant increase in cell numbers of 21% in *amotl2a^{−/−}* mutants as compared to their siblings (*Figure 4I*). In addition and as in morphants, pLLP migrated slower

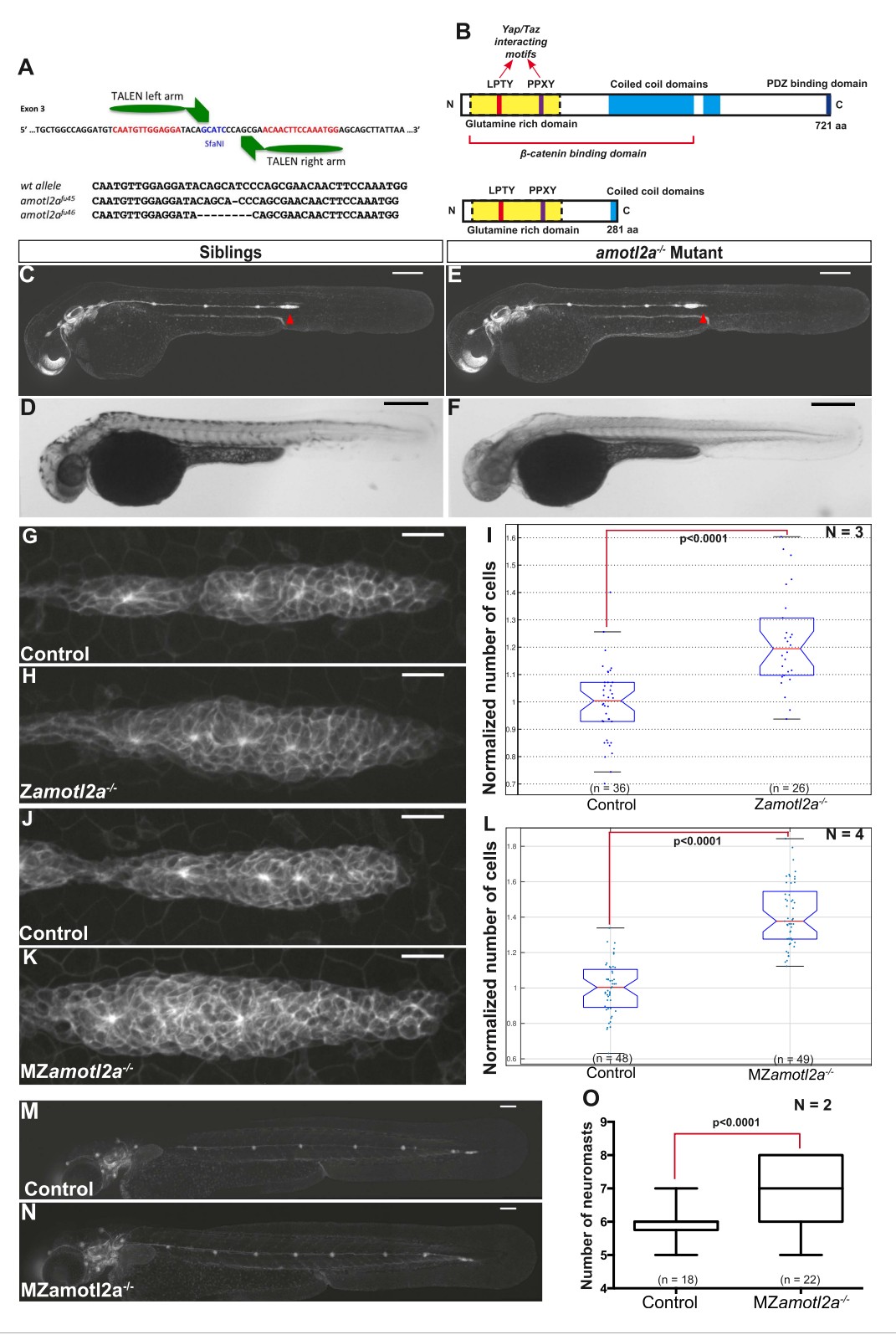

**Figure 4**. *amotl2a* mutants phenocopy the morphant phenotype. (**A**) Scheme showing the transcription activator-like effector nuclease (TALEN) target site in the *amotl2a* locus with the left and right TALEN-binding sites in red separated by the spacer including the restriction site used for screening (blue) (top). Alignment of the two conserved *amotl2a* mutant alleles with the corresponding wild-type sequence showing the deleted nucleotides (bottom). *Figure 4. continued on next page*

*Figure 4. Continued*

(**B**) Scheme comparing the functional domains present in the wild-type Amotl2a protein (721aa long) and the putative truncated proteins (272aa +17 or +9 missense aa for allele *fu45* and *fu46*, respectively). (**C–F**) 36 hpf *cldnb:gfp* wild-type sibling (**C**, **D**) or *amotl2a*$^{-/-}$ mutant embryo (**E**, **F**) imaged with fluorescent (**C**, **E**) or transmitted light (**D**, **F**). (**G**, **H**, **J**, **K**) MIP of Z-stacks of pLLP in *cldnb:gfp* embryos with the indicated genetic background. (**I**, **L**) Boxplots comparing the cell numbers between the indicated genetic backgrounds. (**M**, **N**) MZ*amotl2a*$^{-/-}$ mutant embryo (**N**) showing an extra deposited neuromast as compared to a wild-type sibling embryo (**M**). (**O**) Boxplot showing the corresponding quantification (*Figure 4—source data 1, 2*; *Figure 4—figure supplements 1, 2*, *Figure 4—source data 3, 4*).

The following source data and figure supplements are available for figure 4:

**Source data 1**. Cell counts in zygotic *amotl2a* mutants.

**Source data 2**. Number of deposited neuromasts in MZ*amotl2a* mutants.

**Source data 3**. Migration speed in MZ*amotl2a* mutants.

**Source data 4**. Cell counts in neuromasts of morphants (Excel sheet 1 related to panel A) and MZ*amotl2a* mutants (Excel sheet 2 related to panel B).

**Figure supplement 1**. MZ*amotl2a*$^{-/-}$ mutants exhibit reduced pLLP migration speed.

**Figure supplement 2**. Deposited neuromasts or interneuromast chains do not contain more cells in *amotl2a* morphants or mutants.

in *amotl2a*$^{-/-}$ mutants as compared to controls ($-32\%$, p < 0.0001, *Figure 4—figure supplement 1*).

Homozygous mutants were viable, fertile, and generated maternal-zygotic (MZ) mutants that were also morphologically normal, viable, and fertile. Since *amotl2a* is expressed maternally (*Huang et al., 2007*), we wanted to determine whether maternal RNA and/or proteins also contributed to limit the number of cells in the pLLP. pLLP cell counts were further increased to 40% in MZ*amotl2a*$^{-/-}$ mutants as compared to controls (*Figure 4J–L*, p = 1.18E-21), an increase similar to that quantified in *amotl2a* morphants (*Figure 3E,F*).

We then asked where these additional cells ended up later in development. Neither deposited neuromasts nor interneuromast chains comprised more cells at 48 hpf in morphants as compared to controls (*Figure 4—figure supplement 2A*). Similarly, MZ*amotl2a*$^{-/-}$ mutant neuromasts also contained the same number of cells as controls at 72 hpf (*Figure 4—figure supplement 2B*). Yet, there was on average one additional neuromast deposited in morphants (not shown) and mutants (*Figure 4M–O*) as compared to control embryos. Altogether, these results confirmed that Amotl2a is required to limit the number of cells in the pLLP and thus its size. Also, loss of Amotl2a does not affect the size of mature neuromasts but increases their number.

## Zebrafish Amotl2a physically interacts with zebrafish Yap1 and Taz

To dissect the mechanisms by which Amotl2a was limiting the number of cells in the pLLP, we performed a Y2H screen to identify Amotl2a-interacting partners. The screen was performed with a full-length (FL) cDNA of *amotl2a* as bait and a cDNA library of 18–20 hpf zebrafish embryos as prey library (Hybrigenics, France). 25 proteins were identified as potential Amotl2a interaction-partners and were given a score ranging from A (very high confidence in the interaction) to D (moderate confidence in the interaction). The Hippo pathway effectors Yap1 and Taz were among the 10 strongest identified partners with an A and B score, respectively. Further analysis showed that in both cases, the amino acid (aa) sequence shared by all prey fragments (Selected Interaction Domain) included the evolutionarily conserved WW motifs (2 for Yap1 and 1 for Taz, *Figure 5A*).

To further test if the WW motifs were required for the physical interactions, we mutated both and tested the interactions with Amotl2a in a Y2H assay. The Y2H assay first confirmed the interaction

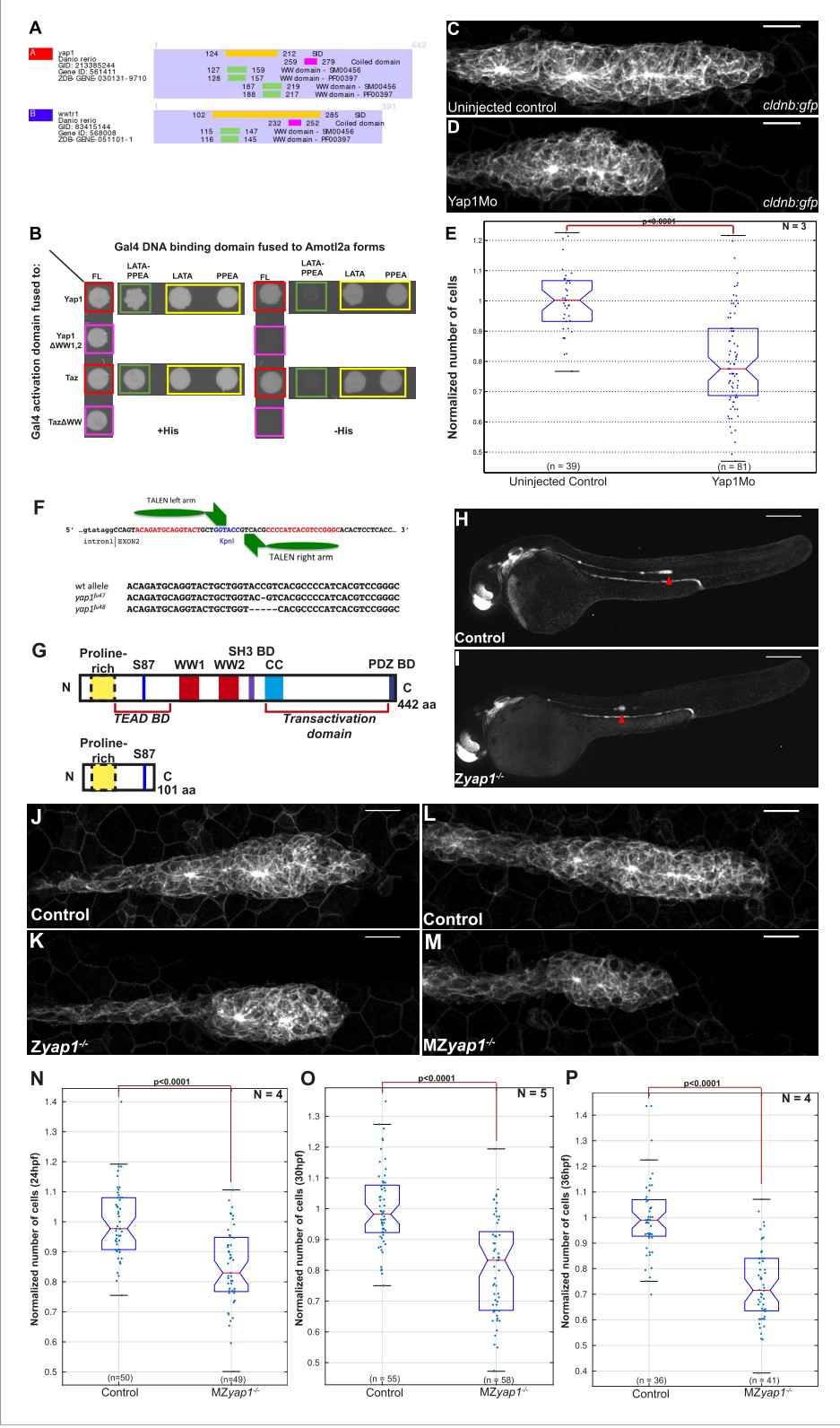

**Figure 5**. Yap1 physically interacts with Amotl2a and is required for the pLLP to have the correct number of cells. (**A**) Part of the yeast two-hybrid (Y2H) screen results showing the Selected Interaction Domain (SID) of Yap1 and Taz (encoded by the gene *wwtr1*) and known functional and structural domain on these proteins. The SID is the amino acid sequence shared by all prey fragments matching the same reference protein, here Yap1 and Taz. (**B**) Y2H assay
*Figure 5. continued on next page*

*Figure 5. Continued*

with Histidine (left panel, growth control) and without Histidine (right panel, protein interaction assay) showing the interactions between zebrafish Amotl2a and Yap1 or Taz (red), but not with the corresponding proteins mutated in the known interaction motifs: LPTY/PPEY for Amotl2a (green) and WW domain for Yap1 and Taz (pink). (**C, D**) MIP of Z-stacks of pLLP in *cldnb:gfp* embryos injected as indicated. (**E**) Boxplot comparing the number of cells in the pLLP in Yap1Mo-injected embryos and controls. (**F**) Scheme showing the TALEN target site in the *yap1* locus with the left and right TALEN-binding sites in red separated by the spacer including the restriction site used for screening (blue) (top). Alignment of the two *yap1* mutant alleles with the corresponding wild-type sequence showing the deleted nucleotides (bottom). (**G**) Scheme comparing the functional domains present in the wild-type Yap1 (442aa long) and the putative truncated proteins (76aa +25 or +45 missense aa before for allele *fu47* and *fu48*, respectively). (**H, I**) Overview pictures of 36 hpf *cldnb:gfp* control and Z*yap1*⁻/⁻ embryos. (**J–M**) MIP of Z-stacks of pLLP in *cldnb:gfp* embryos with the indicated genotype. (**N–P**) Boxplot showing cell counts in the pLLP in embryos of the corresponding genotypes (*Figure 5—source data 1, 2*; *Figure 5—figure supplement 1*; *Figure 5—source data 3, 4*).

The following source data and figure supplement are available for figure 5:

**Source data 1**. Cell counts in *yap1* morphants.

**Source data 2**. Cell counts in MZ*yap1* mutants at 24 hpf (Excel sheet 1 related to panel N), 30 hpf (Excel sheet 2 related to panel O) and 36 hpf (excel sheet 3 related to panel P).

**Source data 3**. Cell counts in Z*yap1* mutants.

**Source data 4**. Number of deposited neuromasts (Excel sheet 1 related to panel D), number of cells (Excel sheet 2 related to panel E), and number of hair cells (Excel sheet 3 related to panel H) per neuromasts in MZ*yap1* mutants.

**Figure supplement 1**. *yap1* and *taz* are ubiquitously expressed at 30 hpf.

---

between zebrafish Amotl2a and zebrafish Yap1 and Taz (red in *Figure 5B*). Furthermore, mutations of the WW motifs of Yap1 and Taz abolished the corresponding interaction with Amotl2a (pink in *Figure 5B*). Conversely, mutations of the LPTY and PPEY motifs of Amotl2a, but not of either of them alone, abolished the interaction with WT Yap1 and Taz (green and yellow, respectively, in *Figure 5B*). This result indicated that, similar to their human orthologs, zebrafish Yap1 and Taz physically interact with zebrafish Amotl2a and that this interaction is mediated by the WW motifs of Yap1/Taz and the LPTY/PPEY motifs of Amotl2a. We therefore hypothesized that the increase in proliferation upon loss of Amotl2a could result from an upregulation of Yap1/Taz activity.

## Yap1 activity is required for the pLLP to reach its normal size

To test whether an upregulation of Yap1/Taz activity could account for the increased cell number in the pLLP upon loss of Amotl2a function, we first assessed the expression of zebrafish *yap1* and *taz*. Both genes were broadly expressed during development including in the pLLP (*Figure 5—figure supplement 1A,B*). We then asked whether Yap1 or Taz was implicated in controlling pLLP cell number using morpholino-based knockdown. Injection of a previously published translation-blocking Mo against Taz (*Hong et al., 2005*) did not cause obvious differences in pLLP size (data not shown). In contrast, embryos injected with a previously published, splice-blocking Yap1 morpholino (Yap1Mo) (*Skouloudaki et al., 2009*; *Fukui et al., 2014*) were morphologically normal but had smaller pLLP than controls and often a rounder shape (*Figure 5C,D*). Cell counts indicated that the pLLP of *yap1* morphant embryos had about 15% less cells than uninjected controls (p = 2.65E-12) (*Figure 5E*). Yet, they completed migration and deposited neuromasts. In order to confirm this essential role of Yap1 in controlling cell number in the pLLP, we generated *yap1* mutants using TALEN-mediated mutagenesis. Like for *amotl2a*, we further analyzed two *yap1* alleles with a frame shift leading to a premature STOP codon in the second exon (*Figure 5F*). The putative resulting truncated proteins would lack most of their functional domains including the Transcriptional Enhancer Activator Domain (TEAD) binding domain, the 2 WW motifs and the PDZ-binding domains (*Figure 5G*). Both alleles had identical

phenotypes. $yap1^{-/-}$ homozygous mutants embryos were morphologically normal (*Figure 5H,I*) with smaller and at times rounder pLLP (*Figure 5J,K*), a phenotype identical to that of the *yap1* morphants. Cell counts confirmed that $yap1^{-/-}$ pLLP also showed a decrease in cell number of 19% (p = 1.25E-05) (*Figure 5—figure supplement 1C*). Interestingly, these embryos again developed into normal looking and fertile adult fish. MZ$yap1^{-/-}$ mutants also developed normally and were morphologically indistinguishable from controls. To determine at which stage Yap1 activity was required in the pLLP, we performed cell counts in MZ$yap1^{-/-}$ mutants when the pLLP starts to migrate (24 hpf), when it reaches the middle of the yolk extension (30 hpf) (*Figure 5L,M*), and at the end of the yolk extension (36 hpf). At 24 hpf, the pLLP of MZ$yap1^{-/-}$ mutants was already 15% smaller than related controls (p = 4.12E-08). This difference continued to increase with time to reach 21% at 30 hpf (p = 2.15E-10) and 27% at 36 hpf (p = 2.41E-14) (*Figure 5N–P*). These results showed that Yap1 is required both before and during migration to establish the correct number of cells in the pLLP.

A recent study reported that *yap1* morphants show a reduced number of neuromasts (*Loh et al., 2014*). Given the reduced number of cells in the pLLP of *yap1* morphants and mutants, we expected a similar result. Intriguingly, neither our morphants (not shown) nor our MZ$yap1^{-/-}$ mutants showed a decreased number of neuromasts (*Figure 5—figure supplement 1D*). Furthermore, these neuromasts did not contain fewer cells (*Figure 5—figure supplement 1E*). These results strongly suggested that compensatory mechanisms kick-in in the pLLP of $yap1^{-/-}$ mutants to allow them to migrate to the tip of the tail while depositing a normal number of neuromasts consisting of a normal number of cells despite their smaller size. It was also reported that neuromasts of *yap1* morphants contain fewer hair cells due to loss of *prox1* expression in these embryos (*Li et al., 2012*). Intriguingly, neither the number of hair cells per neuromast (*Figure 5—figure supplement 1F–H*) nor the expression of *prox1* in the pLLP (*Figure 5—figure supplement 1I–L'*) was affected in our MZ$yap1^{-/-}$ mutants. These discrepancies are discussed below. Altogether, our results indicated that Yap1 activity is required for the pLLP to reach its normal cell number and size. This further supported the idea that the increase in proliferation upon loss of Amotl2a activity could be due to an increase of Yap1 activity.

## Amotl2a inhibits Yap1 activity to limit proliferation in the pLLP

We reasoned that if an increase of Yap1 activity was the cause of the hyperplasia in the pLLP of *amotl2a* morphants and mutants, we should be able to suppress this phenotype by reducing Yap1 levels. The pLLP size of embryos co-injected with Yap1Mo and Amotl2aMo was indeed smaller than that of *amotl2a* morphants alone and was comparable to that of control embryos (*Figure 6A–D*). Automated cell counts revealed that the number of cells in the pLLP of *amotl2a;yap1* double morphants was significantly reduced as compared to *amotl2a* morphants (34% reduction, p = 1.53E-16) and was not significantly different from controls (p = 0.098) (*Figure 6E*). To confirm this result, we generated MZ$amotl2a^{-/-}$;MZ$yap1^{-/-}$ double mutants. These mutants were morphologically indistinguishable from related controls. Here also, the increase in pLLP size in MZ$amotl2a^{-/-}$ was suppressed in the MZ double mutant embryos (*Figure 6—figure supplement 1A–E*). These observations were confirmed by automated cell counts: the 37% increase in MZ$amotl2a^{-/-}$ pLLP (p = 4.62E-11) was suppressed in MZ$amotl2a^{-/-}$;MZ$yap1^{-/-}$ double mutants and comparable to controls (p = 0.06). MZ$yap1^{-/-}$ mutants were, however, still lower than double mutants (−27%, p = 1.09E-11) (*Figure 6—figure supplement 1E*). Together, these results demonstrated that the increased pLLP cell counts that resulted from the loss of Amotl2a function could be suppressed by an additional loss of Yap1 activity. This strongly suggested that the overproliferation in $amotl2a^{-/-}$ mutants pLLP was in part mediated by Yap1.

Therefore, we directly tested if the increased proliferation rate of *amolt2a* morphants was similarly suppressed in double morphants. As previously observed, the proliferation rate in *amotl2a* morphants was significantly increased in the trailing region as compared to controls (control vs Amotl2aMo: 0.27 ± 0.01 and 0.40 ± 0.01, p < 0.0001); and this increase was partially suppressed in embryos co-injected with Amotl2aMo and Yap1Mo (0.32 ± 0.02, p = 0.003) (*Figure 6F–K*). Altogether, these results strongly suggest that Amotl2a is limiting proliferation and cell number in the pLLP, at least in part, by inhibiting Yap1 activity.

Finally, we wanted to test whether the physical interaction between Amotl2a and Yap1 was necessary for Amotl2a function. For this purpose, we attempted to rescue the *amotl2a* morphant phenotype with *amotl2a^m* (wild-type *amotl2a* modified to be insensitive to the Mo) and with

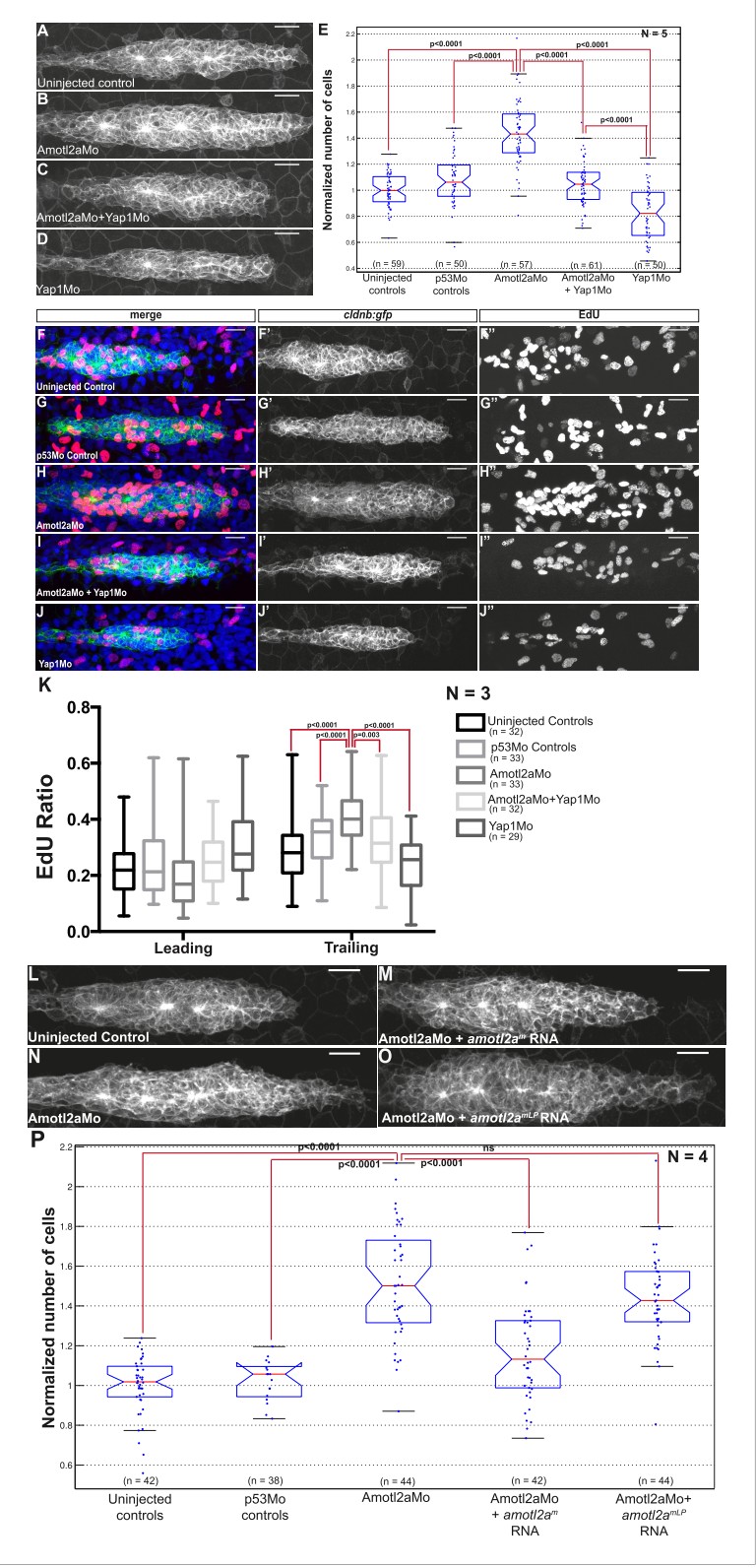

**Figure 6**. Loss of Yap1 suppresses the increased cell proliferation in *amotl2a* morphant. (**A–D**, **L–O**) MIP of Z-stacks of pLLP in *cldnb:gfp* embryos injected as indicated. (**E**, **P**) Corresponding boxplots comparing pLLP cell counts. (**F–J''**) MIP of Z-stacks of pLLP in *cldnb:gfp* embryos stained with EdU and DAPI showing the EdU (right panels), cell membranes (middle panels), and merged channels (left panels). (**K**) Corresponding boxplot showing the EdU index

*Figure 6. continued on next page*

*Figure 6. Continued*

in the leading and trailing regions of the pLLP in the indicated experimental conditions (*Figure 6—source data 1–3*; *Figure 6—figure supplement 1*, *Figure 6—source data 4*).

The following source data and figure supplement are available for figure 6:

**Source data 1**. Cell counts in *amotl2a;yap1* double morphants.

**Source data 2**. EdU ratio in the leading region and trailing region of *amotl2a;yap1* double morphants.

**Source data 3**. Cell counts in embryos co-injected with Amotl2aMo and *amotl2a^m* or with Amotl2aMo and *amotl2a^{mLP}*.

**Source data 4**. Cell counts in MZ*amotl2a*;MZ*yap1* double mutants.

**Figure supplement 1**. Loss of Yap1 in MZ*amotl2a*^{−/−};MZ*yap1*^{−/−} embryos supresses increased pLLP cell number resulting from loss of Amotl2a in MZ*amotl2a*^{−/−} embryos.

*amotl2a^{mLP}*, which was mutated in the LPTY and PPEY motifs and thus could not interact with Yap1. While *amotl2a^m* could partially rescue the increase in cell counts of *amotl2a* morphants as shown above (p = 6.36E-08), *amotl2a^{mLP}* could not (p = 0.18) (*Figure 6L–P*). We also checked that the stability of the two proteins were comparable. Since there was no antibody recognizing zebrafish Amotl2a, we took advantage of the myc-tagged versions of the proteins that we used in our Y2H assays. The two proteins had a comparable stability in yeast (*Figure 6—figure supplement 1F*). Altogether, these results suggested that the physical interaction between Amotl2a and Yap1 is required for the proliferation-limiting activity of Amotl2a.

## Amotl2a suppresses Lef1 activity to limit proliferation in the pLLP

Our results so far indicated that the loss of Yap1 suppressed the increase in cell number in *amotl2a* mutants, possibly via a physical interaction between Amotl2a and Yap1. Yet, pLLP cell counts in MZ*amotl2a*^{−/−};MZ*yap1*^{−/−} double mutants were still higher than in MZ*yap1*^{−/−} suggesting that additional factors mediated the hyperplasia in *amotl2a* mutant pLLPs. Given that Wnt/β-catenin signaling, via Lef1, promotes proliferation in the leading region of the pLLP (*Gamba et al., 2010*; *Aman et al., 2011*; *McGraw et al., 2011*; *Valdivia et al., 2011*; *Matsuda et al., 2013*) and that Amotl2a has been shown to physically interact with β-catenin and inhibit Wnt/β-catenin signaling during zebrafish gastrulation (*Li et al., 2012*), we tested if Wnt/β-catenin signaling could also mediate the increase in cell number upon loss of Amotl2a activity.

We first injected Amotl2aMo in *tg(7xTCFXla.Siam:nlsmCherry)^{ia5}* embryos that carry a Wnt reporter (*Valdivia et al., 2011*; *Moro et al., 2012*). Although the intensity of red fluorescence was not significantly increased upon Amotl2aMo injection (*Figure 7—figure supplement 1A–E*), we could not determine whether the number of cells expressing the reporter was increased because the red fluorescence was already present in most of the pLLP cells in control embryos (*Figure 7—figure supplement 1C,D*), probably due to the stability of the cherry protein. Next, we tested whether the expression of a *bona fide* Wnt/β-catenin transcriptional target, such as *lef1*, was increased upon loss of Amotl2a activity. Both *amotl2a* morphants and mutant embryos showed a slight but significant expansion of the *lef1* expression domain into the trailing region of the primordium as compared to their respective controls (*Figure 7A–E* and *Figure 7—figure supplement 1F–J*). On the other hand, the expression of *lef1* was not modified in MZ*yap1*^{−/−} mutants (*Figure 7—figure supplement 1K–N*). This suggested that an increase in Wnt/β-catenin/Lef1 activity in the trailing region of the pLLP could also partially mediate the increase in cell number. We thus tested if suppressing Lef1 activity in *amotl2a* mutants would suppress this phenotype. For this purpose, we injected MZ*amotl2a*^{−/−} mutant embryos with a previously published *lef1* morpholino (Lef1Mo) and performed cell counts at mid-migration (*Valdivia et al., 2011*). While MZ*amotl2a*^{−/−} mutants showed bigger primordia as expected (36%, p = 2.89E-09) (*Figure 7G*, see 'a' in N), *lef1* morphants at this stage did not show a significant reduction in pLLP cell counts (p = 0.1) (*Figure 7F,J* and 'b' in N). In contrast, cell counts in

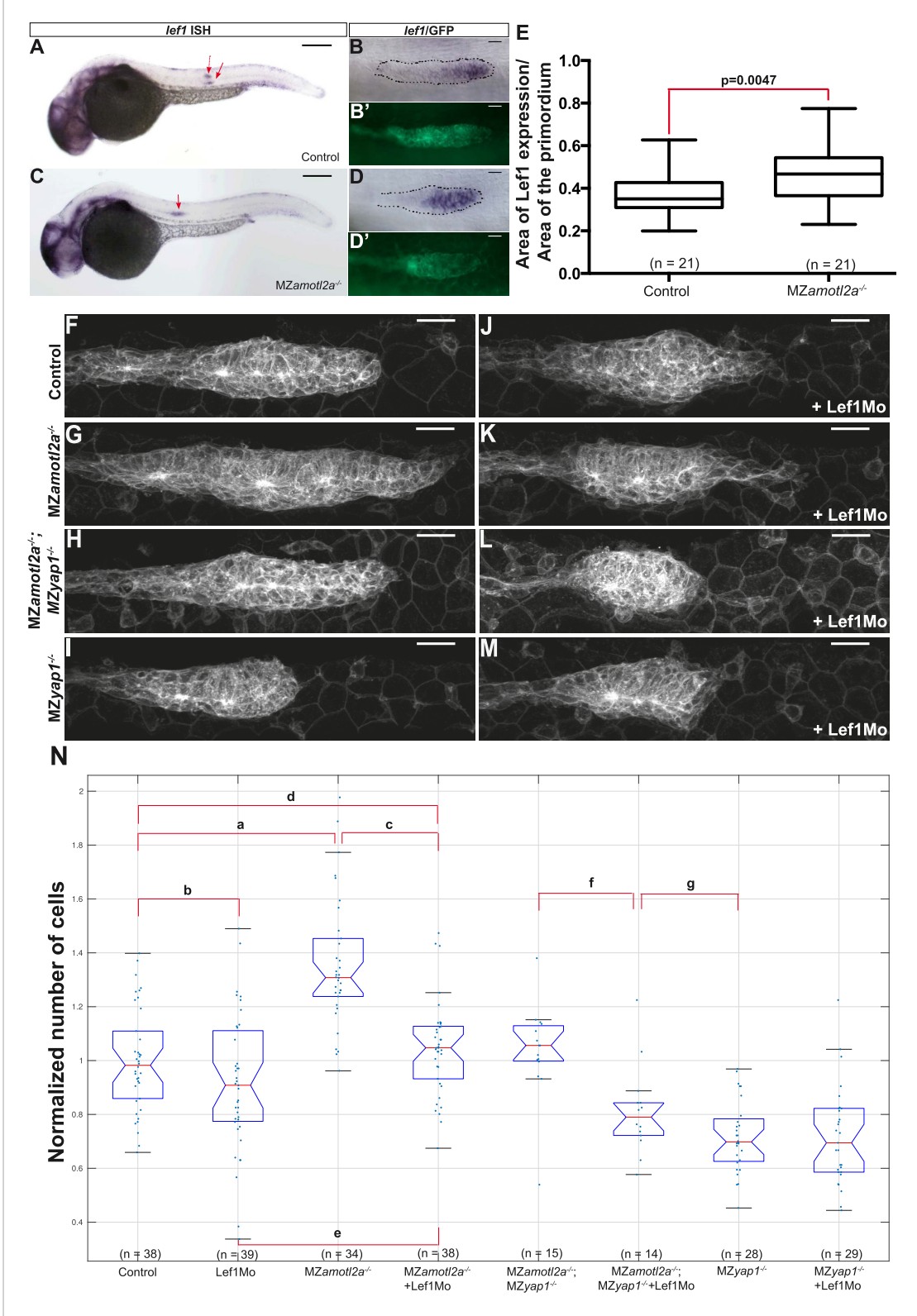

**Figure 7**. Loss of Lef1 suppresses the increased cell proliferation in *amotl2a* mutants. (**A–D′**) 30 hpf *cldnb:gfp* embryos with the indicated genetic background stained with a *lef1* ISH probe and an anti-GFP antibody (**B′**, **D′**). (**E**) Boxplot showing the expansion of *lef1* expression domain in MZ*amotl2a*⁻/⁻ embryos. (**F–M**) MIP of Z-stacks of pLLP in *cldnb:gfp* embryos with the indicated genotype, either uninjected (**F–I**) or injected with a Lef1Mo (**J–M**). (**N**) Corresponding boxplot comparing pLLP cell counts (*Figure 7—source data 1, 2*; *Figure 7—figure supplement 1*, *Figure 7—source data 3*).

*Figure 7. continued on next page*

*Figure 7. Continued*

The following source data and figure supplement are available for figure 7:

**Source data 1**. Relative *lef1* expression domain area in MZ*amotl2a* mutants.
**Source data 2**. Cell counts in Lef1Mo-injected MZ*amotl2a*;MZ*yap1* double mutants.
**Source data 3**. Relative *lef1* expression domain area in *amotl2a* morphants.
**Figure supplement 1**. Loss of Amotl2a leads to an expansion of *lef1* expression.

Lef1Mo-injected MZ*amotl2a*$^{-/-}$ mutants were significantly reduced as compared to MZ*amotl2a*$^{-/-}$ mutants (−23%, p = 5.32E-08, *Figure 7N* 'c') and equivalent to controls (p = 0.28, *Figure 7N* 'd'). Yet, they were still slightly higher than *lef1* morphants (15%, p = 0.01) (*Figure 7G,K* and 'e' in N). These results suggested that the increased cell number in *amotl2a*$^{-/-}$ mutants was Lef1-dependent. To confirm this, we tested if the difference in cell number in MZ*amotl2a*$^{-/-}$;MZ*yap1*$^{-/-}$ and MZ*yap1*$^{-/-}$(see above) was due to Lef1 activity. Indeed when we injected MZ*amotl2a*$^{-/-}$;MZ*yap1*$^{-/-}$ with Lef1Mo, the number of cells in the pLLP was significantly lower than in MZ*amotl2a*$^{-/-}$;MZ*yap1*$^{-/-}$ double mutants (−19%, p = 0.001, *Figure 7N* 'f') and was not significantly different from that of MZ*yap1*$^{-/-}$ mutants ('g' in *Figure 7N*, p = 0.07). Altogether, these results strongly suggested that Amotl2a limits the number of cells and thus the size of the pLLP by repressing both Yap1 and Lef1 proliferation-promoting activities (*Figure 8*), possibly by physically interacting with Yap1 (our data) and β-catenin, as previously shown in zebrafish (*Li et al., 2012*).

## Discussion

In the present study, we describe a novel mechanism underlying control of tissue size in developing vertebrate embryos. We show that the cell junction-associated Motin protein Amotl2a is required to limit proliferation in the pLLP and thus to control the size of this tissue during migration. We further show that hyperplastic pLLP do not generate bigger neuromasts but more neuromasts. Our results indicate that Amotl2a exerts its function by negatively regulating both the Hippo pathway effector Yap1 and the Wnt/β-catenin effector Lef1. Our data point to a new mechanism by which Motin proteins limit tissue size by repressing the Wnt/β-catenin pathway. In addition, while Motin proteins are known to physically interact with, and inhibit YAP/TAZ, this is to our knowledge the first in vivo report showing that Motins control tissue size by interacting with the Hippo pathway effector Yap1 in a living organism.

### Control of proliferation and tissue size in the zebrafish pLLP

Embryonic development is associated with high proliferation rates ensuring proper organ formation. Yet, proliferation must remain under tight spatiotemporal control to ensure that a sufficient number of cells is produced, but not exceeded. In the pLLP, Wnt/βcatenin signaling is required to maintain proliferating progenitors in the leading region to compensate for the loss of cells resulting from neuromast deposition (*Gamba et al., 2010*; *Aman et al., 2011*). This is essential for the pLLP to deposit a complete set of LL organs (*McGraw et al., 2011*; *Valdivia et al., 2011*). However, no mechanism has been proposed that restricts proliferation and growth in the pLLP. We show here that Amotl2a is essential to limit proliferation of the cells that assemble into proneuromasts in the trailing region of the pLLP. This leads us to propose that the size of the pLLP is controlled throughout migration by a balance between Wnt/βcatenin-dependent mitogenic signals from the leading region and Amotl2a-dependent proliferation-restricting signals from the trailing region (*Figure 8*).

We further show that *amotl2a* is expressed downstream of FGF signaling in the trailing region of the pLLP. Yet, embryos lacking FGF signaling in the pLLP do not have larger primordia (*Aman et al., 2011*, and our own observations). This suggests that FGF has a mitogenic function in the pLLP in addition to promoting *amotl2a* expression, and thus, limiting proliferation. This is not surprising given that FGF signaling promotes proliferation in several contexts during vertebrate development

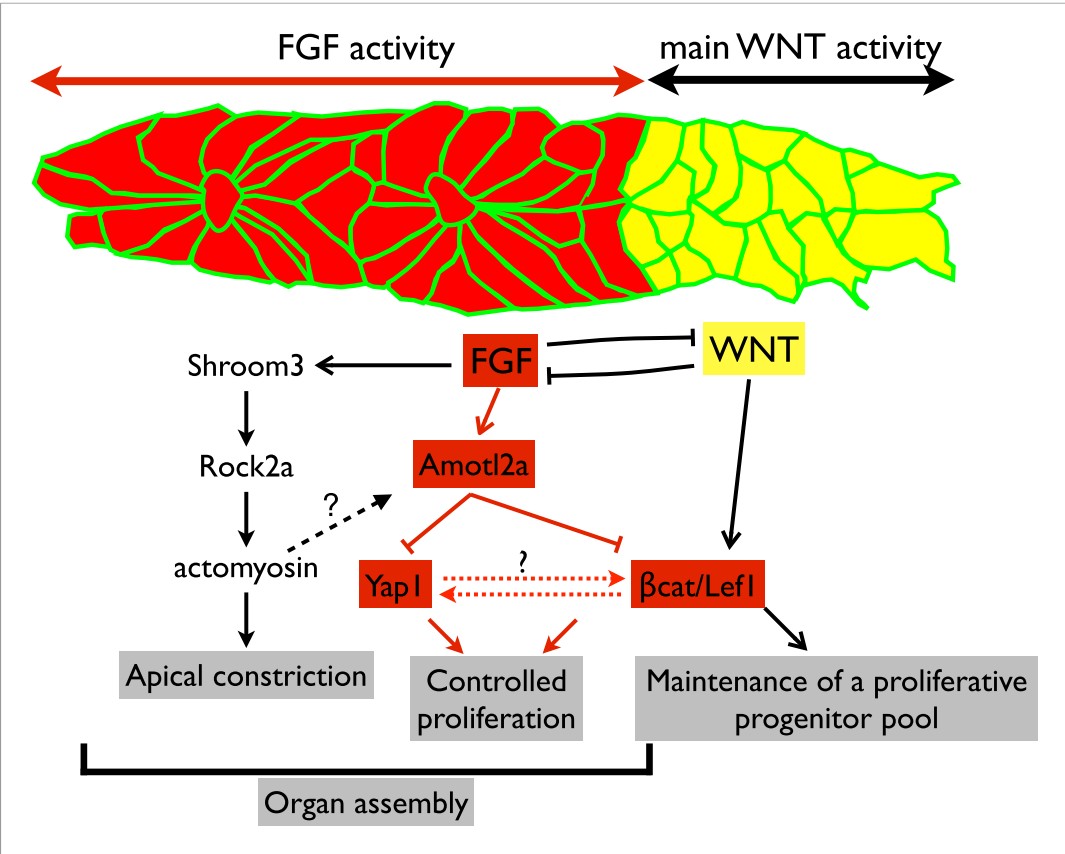

**Figure 8**. Working model. Scheme summarizing the main regulatory pathways involved in proliferation control and organ formation in the pLLP. The red part corresponds to new findings from the present study. Dotted lines indicate hypothetical links.

(*Thisse and Thisse, 2005*; *Aman et al., 2011*). We propose that in FGF-deficient embryos, an increase in Yap1 and Lef1 activity resulting from decreased *amotl2a* expression might partially compensate for the absence of FGF-dependent proliferation.

The increase in cell number in the pLLP of *amotl2a* mutants does not lead to bigger neuromasts, but to one additional neuromast. While a recent study reported that the size of differentiated neuromasts depends on the canonical Wnt pathway at later stages (*Wada et al., 2013*), the mechanisms that dictate how many cells participate in forming a new rosette in the migrating pLLP are not known. Intriguingly, our results suggest that this number does not depend on the size of the pLLP itself. This is in agreement with the fact that smaller primordia, for example in *lef1* mutants, also deposit fewer, but apparently not smaller, neuromasts (*McGraw et al., 2011*; *Valdivia et al., 2011*).

Finally, we show that rosette assembly is slightly delayed in *amotl2a* morphants and mutants. At least two mechanisms could account for this delay. First, Motin proteins have been implicated in the formation of tight junctions in different contexts (*Bratt et al., 2002*; *Sugihara-Mizuno et al., 2007*; *Zheng et al., 2009*), although tight junctions do form in the absence of Amotl2a, their assembly might be challenged, leading to a delay in epithelial rosette assembly. Second, we show that *lef1* expression is expanded upon loss of Amotl2a function. Interestingly, upon loss of Lef1 activity, rosettes assemble closer to the pLLP tip (*McGraw et al., 2011*; *Matsuda et al., 2013*). This has been attributed to the loss of the FGF pathway inhibitor Dusp6 downstream of Lef1 leading to a premature incorporation of leading cells into epithelial rosettes (*McGraw et al., 2011*; *Matsuda et al., 2013*). The expansion of *lef1* expression into the trailing region upon loss of Amotl2a function could conversely explain the delay in rosette assembly. That this delay does not result in an increased distance between deposited neuromasts could be explained by the fact that one additional neuromast form in this context due to the increase in cell number.

## Yap1 mutant pLL primordia are smaller

We show for the first time that the Hippo signaling pathway effector Yap1 is required for the pLLP to contain a sufficient number of cells. Since Yap1 is already required before migration starts, we propose that one of Amotl2a's functions is to maintain low Yap1 activity as rosettes assemble in the trailing region. In *amotl2a* mutants, Yap1 activity would fail to be repressed and would remain higher for a longer period of time leading to overproliferation. Our data do not allow us to determine, however, whether the canonical Hippo pathway is active in the pLLP to additionally regulate Yap1 activity. More work will be required to answer this interesting question.

A paper recently reported that morpholino-mediated knock-down of Yap1 in zebrafish leads to a reduction of the number of neuromasts but not of the size of the pLLP (*Loh et al., 2014*). They also reported a failure of hair cell differentiation that they attributed to a reduction of expression of the Wnt-target gene *prox1*. Intriguingly, both our *yap1* morphants and mutants showed significantly different phenotypes with a reduction of the size of the pLLP but not of the number of neuromasts deposited. In addition, neither *prox1* expression nor the numbers of hair cells were affected in *yap1*$^{-/-}$ mutants. One possible explanation for these discrepancies is that our *yap1* mutation does not affect all *yap1* transcripts. This is, however, unlikely since it lies in the TEAD-binding domain, which is probably essential to Yap1 function. On the other hand, the authors also reported increased apoptosis and activation of the p53 pathway, two processes that have been shown to lead to a reduction in the number of neuromasts (*Aman et al., 2011*). This could explain the discrepancies of phenotypes between the previously reported *yap1* morphants and our newly generated *yap1*$^{-/-}$ mutants.

## How is amotl2a inhibiting Yap1 activity in the pLLP?

Our findings that Amotl2a physically and genetically interacts with the Hippo pathway component Yap1 to control proliferation in the pLLP is in agreement with several studies showing that Motin proteins physically interact with YAP and TAZ, sequester them in the cytoplasm, and inhibit their proliferation-promoting transcriptional activity (*Varelas et al., 2010*; *Chan et al., 2011*; *Wang et al., 2011*; *Zhao et al., 2011*). Our results provide the first evidence that a similar mechanism could be involved in vivo during vertebrate development to control tissue and organ size. Antibodies recognizing Yap1 will be necessary to determine its localization in the pLLP cells. The commercial Yap1 antibodies we have tested did not cross-react with the zebrafish protein. Intriguingly, none of the known YAP/TAZ target genes we looked at, including *ctgfs*, *cyr61*, *myc*, *cyclins*, *and prox1*, showed differences in expression level in *amotl2a* morphants (data not shown).

Alternatively, rather than sequestering Yap1 into the cytoplasm, Amotl2a could promote Yap1 degradation. Motins have indeed been shown to promote the phosphorylation of YAP/TAZ by LATS1/2, which would not only lead to their cytoplasmic retention but also to their degradation (*Paramasivam et al., 2011*). Although there is currently no evidence that the canonical Hippo pathway is active in the pLLP, Amotl2a could be involved in controlling Yap1 stability.

## What is the link between amotl2a, Yap1, and Lef1 in the pLLP?

Our results indicate that Amotl2a limits the number of cells and thus the size of the migrating pLLP by repressing the activity of both Lef1 and Yap1. This is in agreement with the fact that both Lef1 (*Gamba et al., 2010*; *McGraw et al., 2011*; *Valdivia et al., 2011*; *Matsuda et al., 2013*) and Yap1 (this paper) are required for establishing and maintaining the correct number of cells in the pLLP.

To our knowledge, a direct link between Motin proteins and the Wnt/β-catenin pathway has been described only once, with a physical interaction between zebrafish Amotl2a and β-catenin (*Li et al., 2012*). Although we did not detect β-catenin in our Y2H screen nor identified an interaction between Amotl2a and β-catenin in our Y2H assay (not shown), Amotl2a might be physically interacting with β-catenin in vivo in the pLLP cells thereby limiting its access to the nucleus.

Our data suggest that Yap1 and Lef1 function at least in part independently downstream of Amotl2a since the pLLP of both *yap1* single mutants and *lef1* morphants has less cells than *amotl2a*; *yap1* double mutants and Lef1Mo-injected *amotl2a* mutants, respectively. Yet, it is also possible that both pathways are partially interconnected downstream of Amotl2a. Several recent studies have indeed reported cross-regulations between Yap/Taz and Wnt/β-catenin signaling (*Varelas et al., 2010*; *Fish et al., 2011*; *Heallen et al., 2011*; *Azzolin et al., 2012*; *Imajo et al., 2012*; *Barry et al., 2013*; *Azzolin et al., 2014*; *Piccolo et al., 2014*). In particular, Yap1 and β-catenin have been shown to

physically interact and mutually repress each other in the cytoplasm while rather promoting each other's activity in the nucleus. Further work will be required to precisely dissect potential cross-talks between Yap1 and Wnt/β-catenin in the pLLP.

### Is Amotl2a linking proliferation to morphogenesis?

One particularly interesting property of YAP/TAZ is their capacity to sense and respond to changes in biophysical properties of cells including cell density, cell polarity, cell shape, tension forces, and substrate stiffness (*Dupont et al., 2011*; *Halder et al., 2012*; *Aragona et al., 2013*; *Gaspar and Tapon, 2014*; *Rauskolb et al., 2014*). Our results raise the question whether Yap1 responds to such changes in the pLLP, and to which extent the response depends on Amotl2a function. We show that the repressive function of Amotl2a on Yap1 coincides with the assembly of cells into rosettes. Interestingly, these cells undergo a number of changes including epithelialization, acquisition of a columnar shape, and apical constriction via acto-myosin contraction (*Lecaudey et al., 2008*; *Ernst et al., 2012*). In principle, Yap1 could respond to any of these changes and Amotl2a could act as a mediator. First, tight junction-associated Amotl2a in newly formed rosettes could recruit and inhibit Yap1. In addition, Motins, including Amotl2a (*Hultin et al., 2014*), can bind actin and this interaction has been shown to compete with the binding of Motin to YAP (*Ernkvist et al., 2006*; *Chan et al., 2013*; *Dai et al., 2013*; *Gaspar and Tapon, 2014*). It is thus tempting to propose that Amotl2a could link actomyosin-based cell shape changes such as columnarization and/or apical constriction during rosette assembly to Yap1 activity and proliferation.

Finally, at low density in culture, cells are flat and spread and this morphology promotes nuclear YAP accumulation (*Wada et al., 2011*). In contrast at high density, cells are compact and tall and this is associated with cytoplasmic localization of YAP. In the pLLP, cells in the leading region are rather flat while cells in the trailing region, which assemble into rosettes, are more columnar (*Lecaudey et al., 2008*). Thus, in the primordium, more than the cell density, the assembly of cells into rosettes and the associated changes in cell shape could regulate the cytoplasmic localization of Yap1 in an Amotl2a-dependent manner. Such a mechanism would allow to couple proliferation rate to organogenesis. Upon loss of Amotl2a, in contrast, this coupling would be lost so that proneuromasts assemble but proliferation is not restricted, leading to hyperplastic LLP and additional sensory organs.

## Materials and methods

### Zebrafish lines, mRNA, and morpholino injection

Adult zebrafish were maintained under standard conditions and embryos were staged according to *Kimmel et al. (1995)*. Transgenic lines *Tg(−8.0cldnb:lynEGFP)*[zf106] (*cldnb:gfp*), *Tg(hsp70l:dnfgfr1-EGFP)*[pd1], *Tg(hsp70l:fgf3-Myc)*[zf115], *Tg(7xTCF-Xla.Siam:nlsmCherry)*[ia5] and mutant lines *fgf3/lia*[t21142] and *fgf10/dae*[tbvbo] have been described previously (*Herzog et al., 2004*; *Lee et al., 2005*; *Norton et al., 2005*; *Thisse and Thisse, 2005*; *Lecaudey et al., 2008*; *Ernst et al., 2012*; *Moro et al., 2012*). Heat-shock was performed at 39°C for 15 min. Capped mRNAs were transcribed with the SP6 mMessage mMachine Kit (Ambion). Mo are described in the *Supplementary file 1D*. Amotl2aMo was co-injected with a p53Mo to overcome unspecific cell death (*Gerety and Wilkinson, 2011*) and both uninjected and p53Mo-injected embryos were used as controls.

### Cloning of FL cDNA and mutant form of *amotl2a*, *yap1*, and *taz*

FL *amotl2a*, *amotl2a*ΔSTOP, *MoBS_Amotl2a*, *yap1*, *and taz* were amplified by PCR from zebrafish embryo cDNA using the primer pairs indicated in the *Supplementary file 1B*. PCR products were further cloned into pCS2, pCS2-gfp, pCS2-TdT, pGADT7, or pGBKT7 vectors as indicated in the *Supplementary file 1C*.

### TALEN-mediated mutagenesis and screening

The mutant lines *amotl2a*[fu45], *amotl2a*[fu46], *yap1*[fu47], and *yap1*[fu48] were generated using TALEN. The TALE repeat array plasmid kit was a gift from Daniel Voytas and Adam Bogdanove obtained via Addgene (kit #1000000024). TALE repeat arrays were assembled following the Golden Gate TALEN assembly protocol originally described in *Cermak et al. (2011)*, modified by the Voytas lab and

available on the Addgene website (https://www.addgene.org/static/cms/filer_public/98/5a/985a6117-7490-4001-8f6a-24b2cf7b005b/golden_gate_talen_assembly_v7.pdf). Target sites and the corresponding repeat-variable diresidue (RVD) sequences were chosen using the online tool MoJo Hand (http://talendesign.org/) in exon 2 and 3 of *yap1* and *amotl2a* genes, respectively. The array plasmids were fused to the Fok1 endonuclease in the GoldyTALEN backbone. After linearization, mRNAs were transcribed using the T3 mMessage mMachine Kit (Ambion by Life Technologies GmbH, Darmstadt, Germany) according to the manufacturer's instructions. The two mRNAs corresponding to the left and right arms were then mixed in equal quantities and injected into embryos at the one-cell stage. 20–30 embryos at 48 hpf were collected from each clutch and gDNA was extracted. The target sites were amplified using primers generating a 350–600 bp long PCR product (*Supplementary file 1E*). Efficiency of the TALEN pair was estimated by digesting the PCR product with the restriction site present in the spacer of the target site. If a significant amount of uncut PCR product was observed, the rest of the injected embryos were further grown to adulthood. The resulting mosaic adult fish were out-crossed and genomic DNA was prepared from 50 embryos to identify potential mutations using the same PCR and digestion as described above. F1 fish were finally genotyped by fin clipping. The uncut band (carrying the mutation) was further amplified and sent to sequence.

To obtain MZ mutants, heterozygous carriers were incrossed and the progeny were raised to adulthood. Homozygous wild-type and mutants were identified by fin-clipping. In all experiments with MZ mutants, control embryos originate from incrosses of these homozygous wild-type and are thus related to the MZ mutants in a way that they share the same 'grand-parents'.

## Whole-mount ISH and immunohistochemistry

ISH and immunofluorescence staining were performed according to standard procedures (*Lecaudey et al., 2004*). *amotl2a*, *taz*, and *yap1* ISH probes were amplified by PCR from zebrafish embryo cDNA using the primers indicated in the *Supplementary file 1A*. PCR products were cloned into the pGEM-T vector (Promega GmbH, Germany) according to manufacturer's instructions. The probes for *cxcr7b*, *cxcr4b*, *lef1*, and *prox1* were previously published (*Glasgow and Tomarev, 1998*; *Dorsky et al., 1999*; *Thisse and Thisse, 2005*; *Valentin et al., 2007*). The following antibodies were used: rabbit anti-GFP (1:500; Torrey Pines Biolabs, Secaucus, NJ, United States), mouse anti-GFP (1:500; JL8, Takara Bio Europe/Clontech, France), mouse anti-ZO1 (1:500; GmbH), mouse anti-HCS-1 (1:20, HCS-1 was deposited to the DSHB, Iowa city, IA, United States by Corwin, J), and Alexa dye-conjugated antibodies (1:500; Molecular Probes). Rhodamine-phalloidin (Life Technologies GmbH) was used at 1:100.

## EdU (5-ethynyl-2′-deoxyuridine) treatment

28–30 hpf embryos were dechorionated, incubated with 10 mM EdU solution (Click-iT EdU, Life Technologies GmbH) for 20 min on ice. After washing, embryos were incubated at 28.5°C for 1 hr to allow EdU incorporation and treated with the Click-iT reaction solution following manufacturer's instructions.

## Y2H screen and assay

The Y2H screen was performed by Hybrigenics (Paris, France) with the FL coding sequence of zebrafish *amotl2a* (aa 1–721). 80 millions interactions were analyzed. The Y2H assay was performed using the Matchmaker system and the *AH109* yeast strain (Clontech) according to the manufacturer's instructions. *amotl2a* was fused to the Gal4 DNA-binding domain in the pGBKT7 vector. Zebrafish *yap1*, *yap1ΔWW1,2* (WQDP and WLDP motifs mutated to AQDA and ALDA), *taz* and *tazΔWW* (WHDP motif mutated to AHDA) were fused to the Gal4 activation domain in the pGADT7 vector. All constructs fused to the DNA-binding domain were tested for autoactivation. To check for protein interactions, colonies were scraped off the plates, diluted to an $OD_{600}$ of 0.4, patched on selective plates lacking histidine, and grown for 3–4 days.

## Yeast cell lysates and Western blot

Yeast cell lysates were prepared according to the protocol from Kushnirov with some modifications (*Kushnirov, 2000*). Yeast cultures were grown to a logarithmic stage to reach an $OD_{600}$ 0.4–0.8. Cultures of 2 ml were harvested by centrifugation. After discarding the supernatant, the pellets were resuspended in 500 μl of 0.1 M NaOH, transferred to 2 ml eppendorf tubes, and incubated at room temperature for 10 min followed by centrifugation at

14,000 rpm for 2 min at room temperature. After carefully discarding the supernatant, the pellet was resuspended in 1× sample buffer slightly modified from standard Laemmli (25 mM Tris pH6.8, 30% glycerol, 5% Sodium Dodecyl Sulfate (SDS), 1% bromophenol blue, and 5% β-mercaptoethanol) and dissolved by full speed shaking on a Thermoblock set at 37°C for 20–30 min. The samples were then boiled at 95°C for 5 min. The cell debris was removed by centrifugation at 14,000 rpm for 5 min at room temperature. The supernatant was transferred to a fresh eppendorf tube and stored at −20°C until loading.

15 µl of each of the yeast lysates was electrophoresed on a 10% SDS polyacrylamide gel and transferred to polyvinylidene difluoride (PVDF) membrane (Millipore, by Merck Chemicals GmbH, Germany). The proteins were analyzed by Western blotting with the following antibodies:

1. monoclonal mouse antibody 9E10 (anti-c-myc, Covance, by BioLegend, United Kingdom),
2. monoclonal mouse antibody DM1A (anti-α-tubulin, Sigma-Aldrich Chemie GmbH, Germany).

## Live imaging, image analysis, and statistical analysis

Imaging and image analysis were performed as previously described (*Ernst et al., 2012*). Quantifications of data are presented as boxplot in which the central line is the median of the group and the edges of the box are the 25th and 75th percentiles. The black vertical lines extend to the most extreme data points not considered outliers. In all cell count boxplots, each blue dot corresponds to one data point (one primordium). Statistical analyses were performed with GraphPad Prism6 and Matlab using the Welch's t-test. p values are indicated on each figure and/or in the result part. Differences in cell counts between groups are differences between mean values (and not between median values shown in the boxplots) and are expressed in percentage.

## Image enhancement, cell segmentation, and automated cell counting

Automated cell counting was done using the GFP channel of *cldnb:gfp* transgenic embryos (*Figure 3—figure supplement 1A*). To exclude that the delayed migration in *amotl2a* morphants would impact on the pLLP cell number, cell counts were done when the primordium had reached the middle of the yolk extension ('mid-migration', *Figure 2—figure supplement 1A,C*) both in control embryos (about 30 hpf) and in *amotl2a* morphants and mutants.

Because the cell membrane is a plane-like structure in three dimensions, we used anisotropic diffusion (*Weickert, 1998*) to enhance local consistency in structure and suppress noise (*Figure 3—figure supplement 1B*). The diffusion tensor, which enhances plane-like structures, was constructed from the local Hessian matrix. We then computed the smallest eigenvalue of the local Hessian matrices, which is a good indicator of the cell membrane (*Sato et al., 2000*). The cell segmentation was done by watershed segmentation with h-minima transform (*Soille, 2003*) on the eigenvalue image. Finally, very small segments were filtered out: their regions are merged into neighboring segments by applying a second watershed segmentation (*Figure 3—figure supplement 1C*). A number of publicly available tools, for example, 'Icy' (*de Chaumont et al., 2012*) or Matlab programs, can be used for these image-processing steps.

To generate a segmentation mask of the pLLP, the 3D images were smoothed with a small Gaussian kernel. An optimal threshold was estimated for converting the smoothed intensity image to a binary image (e.g., using the 'graythresh' function in Matlab), so that the intraclass variance of the two regions '0' and '1' is minimized. Based on this pLLP segmentation mask (*Figure 3—figure supplement 1D*), we obtained the cell numbers by counting each segment that had more than 75% of its volume inside the mask. Visual monitoring of two experiments validated the accuracy of the segmentation (*Figure 3—figure supplement 1*).

## Acknowledgements

We are grateful to S Goetter for excellent fish care, to S Godberson, K Stöhr and J Wenzel for their help as rotation students, and to the Life Imaging Center. We are indebted to C Pujades, P Blader, W Driever, G Pyrowolakis, J Schweitzer, M Simons, and the Lecaudey lab for critical reading of the manuscript. This work was supported by the Excellence Initiative of the German Federal and State Governments (GRK1104 [to SA] and BIOSS Centre for Biological Signalling Studies EXC 294 [to SE, OR and VL]), the German Research Foundation (DFG-SFB850-A2 to VL), the Ministry of Science, Research and Art of Baden-Wuerttemberg ('Juniorprofessoren-Programm' [to SA], and a Marie Curie reintegration grant [to VL]).

## Additional information

### Funding

| Funder | Grant reference | Author |
| --- | --- | --- |
| Deutsche Forschungsgemeinschaft (DFG) | GRK1104 | Sobhika Agarwala, Virginie Lecaudey |
| European Commission (EC) | PERG-GA-2009-256292 | Virginie Lecaudey |
| Ministerium für Wissenschaft, Forschung und Kunst Baden-Württemberg (MWK) | 7635.521 | Virginie Lecaudey |
| Deutsche Forschungsgemeinschaft (DFG) | SFB850-A2 | Sobhika Agarwala, Virginie Lecaudey |
| Deutsche Forschungsgemeinschaft (DFG) | EXC 294 | Stefan Eimer, Olaf Ronneberger, Virginie Lecaudey |

The funders had no role in study design, data collection and interpretation, or the decision to submit the work for publication.

### Author contributions

SA, Acquisition of data, Analysis and interpretation of data, Drafting or revising the article; SD, Acquisition of data, Analysis and interpretation of data; KL, AB, SE, Analysis and interpretation of data; LG, SL, SK, Acquisition of data; OR, Conception and design; VL, Conception and design, Analysis and interpretation of data, Drafting or revising the article

### Ethics

Animal experimentation: This study was performed in strict accordance with the recommendations of the German Animal Welfare Act (TierSchG) based on the directive 2010/63/UE of the European parlament. All of the animals were handled according to approved institutional animal care at the University of Freiburg and approved by the local government (Regierungspräsidium Freiburg Permit Number: G-09/05).

## Additional files

### Supplementary file

• Supplementary file 1. Supplementary tables A, B, C, D related to the 'Materials and methods' section.

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
