## [Decision Letter]

Thank you for submitting your work entitled “Amotl2a interacts with the Hippo pathway to control tissue size in zebrafish” for peer review at *eLife*. Your submission has been evaluated by Janet Rossant (Senior editor) and three reviewers, one of whom is a member of our Board of Reviewing Editors.

The following individuals responsible for the peer review of your submission have agreed to reveal their identity: Tanya Whitfield (Reviewing Editor) and Ajay Chitnis (peer reviewer).

The reviewers have discussed the reviews with one another and the Reviewing Editor has drafted this decision to help you prepare a revised submission.

As you will see, while Reviewer 2 is positive about the paper, Reviewers 1 and 3 have some significant concerns, which should be addressed in any resubmission and would require re-review.

The authors should pay particular attention to the following points. Please see the full reviews below for reference.

1) The Yap mutant should be tested to show when Yap is required for proliferation in the lateral line primordium.

2) Relevant literature (51) should be cited and discussed.

3) The relationship to Wnt signalling in both mutants should be addressed.

4) Care should be taken not to over interpret the data. In particular, the description of the degree of overproliferation and involvement of the Hippo kinase cascade should be toned down (including in the title and summary). In addition, the conclusions about the physical interaction between Amotl2a and Yap1 should be treated with more caution, unless the authors can demonstrate stability of the mutant Amotl2a protein. Please see comments from Reviewers 1 and 3.

Reviewer #1:

This is an interesting paper that examines control on tissue size in the zebrafish lateral line primordium. The manuscript is well written and the data are all very clear and well controlled, with thorough statistical analysis. All morpholino experiments have been repeated in mutants and give comparable results, so this study is a nice demonstration that morpholinos can be reliable in some contexts. The Discussion is very speculative, and includes various further hypotheses for mechanistic insight (such as sequestration of Yap by Amotl2 in the cytoplasm, relationship between Fgf, Amotl2 and Yap, and changes in cell shape affecting Yap activity) that are not tested.

1) The further data mentioned in the Discussion (that Yap target genes are not affected in *amotl2* morphants, and the different timing requirements for Amotl2 and Yap in proliferation in the primordium), do not fit easily with the authors' model. The authors suggest that Yap is required for proliferation in the primordium at earlier stages. Since they have Yap mutants available, this difference in timing requirement should be investigated and the data shown.

2) The authors have omitted to cite another paper that deals with Yap expression and function in the zebrafish lateral line primordium (Loh et al., Science Reports 2014). There are some findings that are contradictory to the present manuscript, and these should be acknowledged and discussed. Firstly, Loh et al. find a reduced number of neuromasts in the *yap1* morphant, whereas the current manuscript states that this is “not the case” for their *yap1* morphants (in the subsection “Control of proliferation and tissue size in the zebrafish pLLP”, last paragraph). The Loh et al. study also undertook a microarray study with their morphants, and reported changes in Wnt signalling pathway genes, whereas the current manuscript states that Wnt signalling is unaffected in their *amotl2a* morphants, but do not show the data (second paragraph of the aforementioned subsection). In order to address these discrepancies, and given the known importance of Wnt signalling in lateral line proliferation, the authors should show an analysis of Wnt signalling in both the *amotl2a* and *yap1* mutants, and should cite and discuss the Loh et al. study.

3) The overproliferation in the primordium is described in the manuscript as “massive” (in the summary) or “strong” (in the Introduction and subsection “Control of proliferation and tissue size in the zebrafish pLLP”), but this seems to me to be an overstatement. The estimation of cell increase in morphants and MZ mutants is measured as “almost 40%” (in the subsections “Amotl2a is essential to restrict the number of cells in the pLLP, and thus to limit its size” and “TALEN-induced non-sense mutations in *amotl2a* phenocopy the *amotl2a* knock-down phenotype”). As the authors have shown nicely, this is due to an increase in the proliferation index, but assuming there is no cell death, this only corresponds to one extra round of division in 40% of the cells in the primordium by the mid-migration point. The mutant data indicate that there is no difference in cell number per neuromast, and just one extra neuromast in the posterior line after migration (Figure 4). Would it not be more accurate to describe the overproliferation in the primordium as mild?

4) The entire embryo in the images of the *amotl2a* MZ mutant appears to be significantly bigger than the controls (Figure 4). Is this the case, or is this within the natural variation in size within the batch? Would the mutation be expected to have a systemic effect, given the expression pattern of the gene?

Reviewer #2:

The paper by Agarwala et al. describes the role of Amotl2a and the Hippo pathway in regulating the size of the pLLP. The paper is excellent. The experiments are well laid out and logical and the conclusions are well supported by the observations. The paper describes how FGF-dependent *amotl2a* expression in maturing proneuromasts regulates proliferation in a trailing zone to regulate the size of the pLLP. The observations are supported by experiments using both morpholino knock-down and mutants. The authors discuss the various interesting ways in which Amotl2a might conceivably regulate Yap1 function and how Amotl2a might link cellular morphogenesis, tension, shape etc. to proliferation. The authors are appropriately cautious about making strong conclusions about any specific mechanism, leaving these interesting open questions for future studies.

I believe the paper offers insight that will be of great value not only to the community studying the fascinating posterior lateral line system but the broader scientific community that is interested in how growth is regulated in any organ system. I do not have any significant problems with the paper in its current form. I'd like to say that I had the opportunity to review an earlier paper describing this study at an earlier stage of development and the current study addresses all the previous concerns I had to tell a simpler and much more compelling story. I look forward to seeing the paper published.

Reviewer #3:

In this manuscript, Agawala et al. report a new in vivo function of *amotl2a*. The authors claim that this is the first report of involvement of Hippo pathway in size regulation in the LL system and that this report provides the “first in vivo evidence that the Hippo/Motin interaction is essential to limit tissue during development”.

To investigate the in vivo function of Motin proteins, the authors used zebrafish posterior lateral line primodium (pLLP) that allows dynamic cell behavioral analyses for organogenesis. Although most of the loss-of-function analyses were carried out by using morpholinos, they confirmed the phenotype by the TALEN induced mutants of *amotl2a* and YAP. They first showed that *amotl2a* expression is dependent on FGF signaling. They next showed that Amotl2a limits the pLLP size by restricting YAP mediated cell proliferation. Finally, based on the rescue experiment they concluded that restricting cell proliferation is mediated by the physical interaction of Amotl2a and YAP. Whereas the over-proliferation of pPPL in the *amotl2a* mutant is interesting, the mechanisms presented in the manuscript are mostly reported previously, e.g. FGF dependent expression of *amotl2a* and physical interaction of Amotl2a and YAP. Some of the conclusions are overstated. Indeed, there are alternative interpretations for the result of the rescue experiment as detailed below. Furthermore, more in depth analysis taking advantage of the zebrafish in vivo system would be necessary to publish this manuscript in this journal.

1) The authors investigated whether physical interaction of YAP and Amotl2a is required for the Amotl2a function based on the rescue experiment of *amotl2a* morphant by the expression of mutant *amotl2a* without YAP binding sites. From the failure of the rescue the authors concluded: “This result indicated that the physical interaction between Amotl2a and Yap1 is required for the proliferation-limiting activity of Amotl2a”. My concern is that the negative result (failure of rescue) even together with the evidence of the physical interaction does not fully support such a strong statement. The failure of rescue could be due to the instability of the mutant Amotl2a without YAP binding sites. The authors have to check the stability of the mutant Amotl2a protein. If the mutant Amotl2a is not less stable, overexpression of this protein could interfere with the interaction of endogenous Amotl2a and YAP and it might induce the dominant-negative phenotype.

2) In the Abstract they concluded: “We further provide the first in vivo evidence that the Hippo/Motin interaction is essential to limit tissue size during development.” Even if YAP is required for the limitation of pPPL size, it is premature to conclude this without presenting the activation status of the Hippo core kinase signaling. Amotl2a might inhibit YAP nuclear localization without involving the Hippo kinase cascade as previously suggested.

3) There are some too strong statements with “indicated” but not fully supported data. For instance: “This indicated that Amotl2a might participate […]” (in the second paragraph of the subsection “Amotl2a is not essential for tight junction and proneuromast assembly in the pLLP”). The third paragraph, as well as the last, of the Results section also need to be attenuated.

---

## [Author Response]

*1) The Yap mutant should be tested to show when Yap is required for proliferation in the lateral line primordium*.

We have now performed cell counts in our MZ*yap1*^*-/-*^ mutants at three different stages:

a) At the beginning of migration (24hpf, Figure 5);

b) When the pLL primordia reach the middle of the yolk extension (“mid-migration”, around 30hpf, Figure 5). This is the stage we have used for cell counting throughout the study;

c) When the pLL primordia reach the end of the yolk extension (around 36 hpf, Figure 5).

These cell counts show that, as we predicted, the pLLP of MZ*yap1*^*-/-*^ mutants are already smaller at 24 hpf (-15%, 4 independent experiments). Yet, the difference in number of cells continues to increase until 36 hpf (-27%, 3 independent experiments,) suggesting that Yap1 is still required at these later stages.

24 hpf is the earliest stage we can perform cell counts since our automated algorithm uses the membrane GFP signal in the Tg(*cldnb:lyngfp*) transgenic line which starts to be clearly visible only at that stage. The Results section has been extended accordingly.

*2) Relevant literature (*[51]*) should be cited and discussed*.

Our results indicated that Mo-mediated knock-down (splice blocking Mo E2I2) or a non-sense mutation in *yap1* (non-sense mutation in the region encoding the TEAD-binding domain of Yap1) lead to a reduction in size of the pLLP but not to a reduction in the number of neuromasts deposited. These results are in contradiction with a previous report analyzing the role of Yap1 in the lateral line using a different splice-blocking Morpholino (E1I1) (51) reporting no apparent difference in the pLLP size but a reduced number of neuromasts.

To better understand these discrepancies, we have performed further experiments to precisely compare the phenotypes. We have quantified the number of neuromasts in MZ*yap1*^-/-^ mutants and related controls in 2 independent experiments (Figure 5—figure supplement 1) and could confirm that, like in our morphants, there is no difference in the number of deposited neuromasts. We further show that neuromasts contain the same total number of cells and the same number of hair cells in MZ*yap1*^-/-^ and in controls (Figure 5—figure supplement 1). Finally we show that the expression of *prox1* is not affected in MZ*yap1*^-/-^ mutants (Figure 5—figure supplement 1). Altogether, these results are now described in the Results section (subsection “Zebrafish Amotl2a physically interacts with zebrafish Yap1 and Taz”) and discussed (subsection “Control of proliferation and tissue size in the zebrafish pLLP”).

Loh et al. report an absence of hair cell in *yap1* morphants and attribute it to the loss of *prox1* expression. The basis for this is an older paper reporting that Prox1 is required for hair cells differentiation in neuromasts using a Morpholino-based knock-down approach (Pistocchi et al., 2009). Importantly, a *prox1a* mutant was recently generated and published to re-examine its role in zebrafish lymphangiogenesis (van Impel et al., 2014). Although the role of Prox1a in hair cell specification was not examined in this paper, the mutant does not show the lymphatic phenotype previously described using Morpholino-mediated knock-down. Therefore the role of Prox1a in hair cell specification would need to be confirmed in these new mutants. One intriguing result for us of the Loh et al. paper was that they could rescue the loss of hair cells in *yap1* morphant by injecting a *yap1* mRNA. With the relative facility to generate mutants since recently, several examples have been reported of Morpholino phenotype that were considered specific because “rescuable” but still were not present in null mutants (Stainier, Kontarakis and Rossi, 2015; Kok et al., 2015).

*3) The relationship to Wnt signalling in both mutants should be addressed*.

We have performed a number of experiments to dissect the relationship between Amotl2a/Yap1 and Wnt. We would like to especially thank the reviewers for their comments here since these experiments have allowed us to show that the Wnt/β-catenin pathway, via Lef1, partially mediates the increase in cell number upon loss of Amotl2a activity, in addition to Yap1. This further allowed us to explain why losing Yap1 function in *amotl2a* mutants or morphants suppressed the increase in cell number to a control level but not to the level of *yap1* mutants.

First, we could show that there is an expansion of *lef1* expression towards the trailing region in *amotl2a* morphants and mutants (Figure 7 and Figure 7—figure supplement 1), but no modification in MZ*yap1*^*-/-*^ mutants (Figure 7—figure supplement 1). This suggested that Wnt/β-catenin was upregulated in the trailing region in the absence of Amotl2a and pointed to a potential role of Lef1 in mediating the increase in cell number in the pLLP in addition to Yap1.

Second, we tested this hypothesis by knocking-down Lef1 in *amotl2a* mutants and *amotl2a;yap1* double mutants. Knocking-down Lef1 in *amotl2a* mutants suppressed the increase in cell number to wild-type level indicating that Lef1 was required for this increase in cell number (Figure 7). In addition, knocking-down Lef1 in *amotl2a;yap1* double mutants fully suppressed the increase in cell number to the level of the *yap1* mutants (Figure 7) strongly suggesting that both Yap1 and Lef1 are together mediating the increase in cell number upon loss of Amotl2a function.

These results are now described in the Results section and discussed in the Discussion.

*4) Care should be taken not to over interpret the data. In particular, the description of the degree of overproliferation and involvement of the Hippo kinase cascade should be toned down (including in the title and summary). In addition, the conclusions about the physical interaction between Amotl2a and Yap1 should be treated with more caution, unless the authors can demonstrate stability of the mutant Amotl2a protein. Please see comments from Reviewers 1 and 3*.

We modified the text accordingly to tone down our description of the overproliferation phenotype.

We also realized that our use of the term “Hippo pathway” was confusing since we never meant in this case the canonical Hippo kinase cascade but rather one of the non-canonical ways Yap1/Taz can be regulated, i.e. via a Motin protein. Although the “Hippo pathway/signaling” is often used to refer to both the canonical and non-canonical part of the pathway in recent papers and reviews, we realized it brought confusion and tried whenever possible to clarify this by using “Hippo pathway effector Yap1” instead.

Concerning the physical interaction, since we do not have an antibody recognizing zebrafish Amotl2a to directly compare the stability of the WT and the mutated version of the proteins *in vivo*, we compared the stability of the Myc-tagged versions that we used in our Y2H assay by western-blot. This shows that both proteins are similarly stable in yeast, the mutated protein being even slightly more stable (Figure 6—figure supplement 1). Yet, since we could not prove that the proteins are similarly stable in vivo and that we cannot completely exclude a dominant-negative effect of the mutated protein, as suggested by reviewer 3, we toned down this conclusion in the text using “suggested” instead of “indicated” (subsection “Amotl2a inhibits Yap1 activity to limit proliferation in the pLLP”).

Reviewer #1:

*This is an interesting paper that examines control on tissue size in the zebrafish lateral line primordium. The manuscript is well written and the data are all very clear and well controlled, with thorough statistical analysis. All morpholino experiments have been repeated in mutants and give comparable results, so this study is a nice demonstration that morpholinos can be reliable in some contexts. The Discussion is very speculative, and includes various further hypotheses for mechanistic insight (such as sequestration of Yap by Amotl2 in the cytoplasm, relationship between Fgf, Amotl2 and Yap, and changes in cell shape affecting Yap activity) that are not tested*.

We thank reviewer 1 for his positive comments. The Discussion has been modified according to the new results we obtained during this revision period. We tried to remove or tone down some of the speculative discussions.

*1) The further data mentioned in the Discussion (that Yap target genes are not affected in* amotl2 *morphants, and the different timing requirements for Amotl2 and Yap in proliferation in the primordium), do not fit easily with the authors' model. The authors suggest that Yap is required for proliferation in the primordium at earlier stages. Since they have Yap mutants available, this difference in timing requirement should be investigated and the data shown*.

As mentioned previously, we have now performed cell counts in our MZ*yap1*^*-/-*^ mutants at three different stages. Please see our response to the first point of the editor’s concerns.

*2) The authors have omitted to cite another paper that deals with Yap expression and function in the zebrafish lateral line primordium (Loh et al., Science Reports 2014). There are some findings that are contradictory to the present manuscript, and these should be acknowledged and discussed. Firstly, Loh et al. find a reduced number of neuromasts in the* yap1 *morphant, whereas the current manuscript states that this is “not the case” for their* yap1 *morphants (in the subsection “Control of proliferation and tissue size in the zebrafish pLLP”, last paragraph). The Loh et al. study also undertook a microarray study with their morphants, and reported changes in Wnt signalling pathway genes, whereas the current manuscript states that Wnt signalling is unaffected in their* amotl2a *morphants, but do not show the data (second paragraph of the aforementioned subsection). In order to address these discrepancies, and given the known importance of Wnt signalling in lateral line proliferation, the authors should show an analysis of Wnt signalling in both the* amotl2a *and* yap1 *mutants, and should cite and discuss the Loh et al. study*.

Please see our response to the second essential revision requested by the editor.

*3) The overproliferation in the primordium is described in the manuscript as “massive” (in the summary) or “strong” (in the Introduction and subsection “Control of proliferation and tissue size in the zebrafish pLLP”), but this seems to me to be an overstatement. The estimation of cell increase in morphants and MZ mutants is measured as “almost 40%” (in the subsections “Amotl2a is essential to restrict the number of cells in the pLLP, and thus to limit its size” and “TALEN-induced non-sense mutations in* amotl2a *phenocopy the* amotl2a *knock-down phenotype”). As the authors have shown nicely, this is due to an increase in the proliferation index, but assuming there is no cell death, this only corresponds to one extra round of division in 40% of the cells in the primordium by the mid-migration point. The mutant data indicate that there is no difference in cell number per neuromast, and just one extra neuromast in the posterior line after migration (*Figure 4*)*. *Would it not be more accurate to describe the overproliferation in the primordium as mild?*

We modified the text accordingly to tone down our description of the overproliferation phenotype.

*4) The entire embryo in the images of the* amotl2a *MZ mutant appears to be significantly bigger than the controls (*Figure 4*). Is this the case, or is this within the natural variation in size within the batch? Would the mutation be expected to have a systemic effect, given the expression pattern of the gene?*

This is not the case. Both MZ*yap1*^*-/-*^ and MZ*amotl2a*^*-/-*^ embryos are indistinguishable from wild-type in terms of overall shape and size. The former picture in Figure 4 was maybe misleading because part of the embryo tail was not well visible. We have changed this picture. In addition, we have taken low magnification pictures with transmitted light of 3 wild-type, 3 MZ*yap1*^*-/-*^, 3 MZ*amotl2a*^*-/-*^ and 3 MZ*yap1*^*-/-*^;*amotl2a*^*-/-*^ at the same stage mounted next to each other in one petri dish for Reviewer 1 (Figure 9).

Author response image 1.**DOI:**
http://dx.doi.org/10.7554/eLife.08201.045

Reviewer #3:

*1) The authors investigated whether physical interaction of YAP and Amotl2a is required for the Amotl2a function based on the rescue experiment of* amotl2a *morphant by the expression of mutant* amotl2a *without YAP binding sites. From the failure of the rescue the authors concluded: “This result indicated that the physical interaction between Amotl2a and Yap1 is required for the proliferation-limiting activity of Amotl2a”. My concern is that the negative result (failure of rescue) even together with the evidence of the physical interaction does not fully support such a strong statement. The failure of rescue could be due to the instability of the mutant Amotl2a without YAP binding sites. The authors have to check the stability of the mutant Amotl2a protein. If the mutant Amotl2a is not less stable, overexpression of this protein could interfere with the interaction of endogenous Amotl2a and YAP and it might induce the dominant-negative phenotype*.

We thank reviewer 3 for his valuable comments. Since we do not have an antibody recognizing zebrafish Amotl2a to directly compare the stability of the WT and the mutated version of the proteins in vivo, we compared the stability of the Myc-tagged versions that we used in our Y2H assay by western-blot. This shows that both proteins are similarly stable in yeast, the mutated protein being even slightly more stable (Figure 6—figure supplement 1). Yet, since we could not prove that the proteins are similarly stable in vivo and that we cannot completely exclude a dominant-negative effect of the mutated protein, as suggested by Reviewer 3, we toned down this conclusion in the text using “suggested” instead of “indicated” (in the subsection “Amotl2a inhibits Yap1 activity to limit proliferation in the pLLP”).

*2) In the Abstract they concluded, “We further provide the first in vivo evidence that the Hippo/Motin interaction is essential to limit tissue size during development.” Even if YAP is required for the limitation of pPPL size, it is premature to conclude this without presenting the activation status of the Hippo core kinase signaling. Amotl2a might inhibit YAP nuclear localization without involving the Hippo kinase cascade as previously suggested*.

We fully agree with Reviewer 3 on that, and it was not at all our intention to even suggest that the canonical Hippo pathway is playing a role in the pLL system. This made us realize that our use of the term “Hippo pathway” was confusing. We never meant in this case the canonical Hippo kinase cascade but rather one of the several non-canonical paths by which Yap1/Taz can be regulated independent of the canonical Hippo kinase cascade, in our case via a Motin protein.

Although the “Hippo pathway/signaling” is often used to refer to both the canonical and non-canonical part of the pathway in recent papers and reviews, we realized it brought confusion and tried whenever possible to clarify this by using “Hippo pathway effector Yap1” instead of “Hippo pathway alone”, including in the title.

*3) There are some too strong statements with “indicated” but not fully supported data. For instance: “This indicated that Amotl2a might participate […]” (in the second paragraph of the subsection “Amotl2a is not essential for tight junction and proneuromast assembly in the pLLP”). The third paragraph, as well as the last, of the Results section also need to be attenuated*.

We have attenuated these statements:

“This result suggested that rosette assembly is slightly delayed in *amotl2a* morphants but that Amotl2a is not essential for tight junction assembly nor for neuromast formation and deposition” (in the subsection “Amotl2a is not essential for tight junction and proneuromast assembly in the pLLP”).

“Expression of the two G-protein coupled receptors *cxcr4b* and *cxcr7b* were not detectably affected in morphants (Figure 2—figure supplement 2), suggesting that a deregulation of their expression is unlikely to account for the decrease in migration speed” (in the same subsection).

“Altogether these results suggested that while rosette assembly and migration are delayed in *amotl2a* morphants, the pLLP deposits neuromasts and eventually migrates to the tip of the tail” (in the same subsection).

“Altogether, these results suggested that the physical interaction between Amotl2a and Yap1 is required for the proliferation-limiting activity of Amotl2a” (in the subsection “Amotl2a inhibits Yap1 activity to limit proliferation in the pLLP”).

References:

Kok FO, Shin M, Ni C-W, Gupta A, Grosse AS, van Impel A, et al. Reverse Genetic Screening RevealsPoor Correlation between Morpholino-Induced and Mutant Phenotypes in Zebrafish. Dev Cell. Elsevier Inc; 2015 Jan 12;32(1):97–108.

Pistocchi A, Feijóo CG, Cabrera P, Villablanca EJ, Allende ML, Cotelli F. The zebrafish prospero homolog prox1 is required for mechanosensory hair cell differentiation and functionality in the lateral line. BMC Dev Biol. 2009;9:58.

Stainier DYR, Kontarakis Z, Rossi A. Making Sense of Anti-Sense Data. Dev Cell. Elsevier Inc; 2015 Jan 12;32(1):7–8.

van Impel A, Zhao Z, Hermkens DMA, Roukens MG, Fischer JC, Peterson-Maduro J, et al. Divergence of zebrafish and mouse lymphatic cell fate specification pathways. Development. 2014 Mar;141(6):1228–38.